# SLICED WASSERSTEIN ESTIMATION WITH CONTROL VARIATES

**Khai Nguyen & Nhat Ho**
Department of Statistics and Data Sciences
The University of Texas at Austin
Austin, TX 78712, USA
`{khainb,minhnhat}@utexas.edu`

## ABSTRACT

The sliced Wasserstein (SW) distances between two probability measures are defined as the expectation of the Wasserstein distance between two one-dimensional projections of the two measures. The randomness comes from a projecting direction that is used to project the two input measures to one dimension. Due to the intractability of the expectation, Monte Carlo integration is performed to estimate the value of the SW distance. Despite having various variants, there has been no prior work that improves the Monte Carlo estimation scheme for the SW distance in terms of controlling its variance. To bridge the literature on variance reduction and the literature on the SW distance, we propose computationally efficient control variates to reduce the variance of the empirical estimation of the SW distance. The key idea is to first find Gaussian approximations of projected one-dimensional measures, then we utilize the closed-form of the Wasserstein-2 distance between two Gaussian distributions to design the control variates. In particular, we propose using a lower bound and an upper bound of the Wasserstein-2 distance between two fitted Gaussians as two computationally efficient control variates. We empirically show that the proposed control variate estimators can help to reduce the variance considerably when comparing measures over images and point-clouds. Finally, we demonstrate the favorable performance of the proposed control variate estimators in gradient flows to interpolate between two point-clouds and in deep generative modeling on standard image datasets, such as CIFAR10 and CelebA[1].

## 1 INTRODUCTION

Recent machine learning applications have recognized the Wasserstein distance (Villani, 2008b; Peyré & Cuturi, 2019), as a universal effective tool. In particular, there are various applications that achieve notable performance by using the Wasserstein distance as a component, for example, (hierarchical) clustering (Ho et al., 2017), domain adaptation (Courty et al., 2016; Damodaran et al., 2018), generative modeling (Arjovsky et al., 2017; Tolstikhin et al., 2018), and so on. Despite its appealing performance, the Wasserstein distance has two major weaknesses. The first weakness is that it has high computational complexities. As discussed in (Peyré & Cuturi, 2019), the time complexity of computing the Wasserstein distance between two discrete measures that have at most $n$ supports is $\mathcal{O}(n^3 \log n)$. Additionally, the space complexity required for the pair-wise transportation cost matrix is $\mathcal{O}(n^2)$. As a result, computing the Wasserstein distance on a large-scale discrete distribution is expensive. The second weakness is that the Wasserstein distance suffers from the curse of dimensionality. More specifically, the sample complexity of the Wasserstein distance is $\mathcal{O}(n^{-1/d})$. Therefore, using the Wasserstein distance in high-dimensional statistical inference may not be stable.

The Wasserstein distance has a famous alternative called the sliced Wasserstein (SW) distance, which is derived from the one-dimensional Wasserstein distance as its base metric. The one-dimensional Wasserstein distance has a closed-form solution, which can be computed in $\mathcal{O}(n \log n)$ time complexity and $\mathcal{O}(n)$ space complexity in the discrete case. Here, "supports" refers to the discrete probability measures being compared. To apply this closed-form solution in a high-dimension setting, random

---

[1]Code for the paper is published at `https://github.com/khainb/CV-SW`.

projections are employed, transforming two measures into infinite pairs of one-dimensional measures using a projecting function with infinite projecting directions that belong to the unit-hypersphere. The closed-form of the one-dimensional Wasserstein distance is then applied to pairs of one-dimensional measures to obtain projections, and the SW distance is computed by aggregating the values of projections. The most commonly used method of aggregation is averaging, resulting in an expectation of the projected one-dimensional Wasserstein distance with respect to the uniform distribution over projecting directions. Since the space of projecting directions is the unit-hypersphere, the SW distance does not suffer from the curse of dimensionality, and its sample complexity is $\mathcal{O}(n^{-1/2})$. Due to its scalability, the SW distance has been successfully applied in various applications such as point-cloud reconstruction (Nguyen et al., 2023; Savkin et al., 2022), generative models (Deshpande et al., 2018), domain adaptation (Lee et al., 2019), shape matching (Le et al., 2024b), clustering (Kolouri et al., 2018), variational inference (Yi & Liu, 2021), and many other tasks.

The SW distance is typically estimated using the Monte Carlo integration, as the intractable expectation requires approximation in practice. The number of Monte Carlo samples drawn from the unit-form distribution over the unit-hypersphere is referred to as the number of projections. Increasing the number of projections improves the accuracy and stability of the estimation. However, to the best of our knowledge, there is no prior work on improving the Monte Carlo approximation scheme of the SW distance, especially in terms of variance reduction. In this work, we propose a novel approach that combines the SW distance literature with the variance reduction literature by introducing the very first control variates, which are a technique for reducing the variance of Monte Carlo estimates.

**Contribution.** In summary, our contributions are two-fold: 1. To address the issue of variance in estimating the SW distance using Monte Carlo samples, we propose a novel approach based on control variates. Specifically, we first identify two Gaussian distributions that closely approximate the two projected one-dimensional measures by minimizing Kullback–Leibler divergence. Next, we construct control variates using the closed-form solution of the Wasserstein distance between two Gaussian distributions. We propose two control variates, an upper bound and a lower bound, for the Wasserstein distance between the fitted Gaussian distributions. By using these control variates, we show that the computation is only linear in terms of the number of supports and the number of dimensions, even when dealing with discrete probability measures. This means that the proposed control variates estimators have the same computational complexity as the conventional estimator of the SW distance. Overall, our proposed approach provides a practical and efficient method for estimating the SW distance with reduced variance.

2. We empirically show that the proposed control variate estimators yield considerably smaller variances than the conventional computational estimator of SW distance, using a finite number of projections, when comparing empirical distributions over images and point clouds. Moreover, we illustrate that the computation for control variates is negligible compared to the computation of the one-dimensional Wasserstein distance. Finally, we further demonstrate the favorable performance of the control variate approach in gradient flows between two point clouds, and in learning deep generative models on the CIFAR10 (Krizhevsky et al., 2009) and CelebA (Liu et al., 2015) datasets. From our experiments, we observe that using the proposed control variate estimators considerably improves performance, while their computation time is approximately the same as that of the conventional estimator.

**Organization.** The remainder of the paper is organized as follows. First, we review the background on the Wasserstein distance, including its special cases, such as the one-dimensional case and the Gaussian case, as well as the sliced Wasserstein distance and the Monte Carlo estimation scheme, in Section 2. Next, we discuss the construction of control variates and their resulting estimator of the SW distance in Section 3. We then present experimental results in Section 4 to demonstrate the benefits of the proposed control variate estimators. Finally, we conclude in Section 5 and provide proofs of key results, related works, and additional materials in the Appendices.

**Notations.** For any $d \geq 2$, we denote $\mathbb{S}^{d-1} := \{\theta \in \mathbb{R}^d \mid ||\theta||_2^2 = 1\}$ as the unit hyper-sphere and $\mathcal{U}(\mathbb{S}^{d-1})$ as its corresponding uniform distribution. We denote $\mathcal{P}(\mathbb{S}^{d-1})$ as the set of all probability measures on $\mathbb{S}^{d-1}$. For $p \geq 1$, $\mathcal{P}_p(\mathbb{R}^d)$ is the set of all probability measures on $\mathbb{R}^d$ that have finite $p$-moments. For any two sequences $a_n$ and $b_n$, the notation $a_n = \mathcal{O}(b_n)$ means that $a_n \leq Cb_n$ for all $n \geq 1$, where $C$ is some universal constant. We denote $\theta \sharp \mu$ as the push-forward measure of $\mu$ through the function $f : \mathbb{R}^d \to \mathbb{R}$, where $f(x) = \theta^\top x$. We denote $P_X$ as the empirical measure $\frac{1}{m} \sum_{i=1}^{m} \delta_{x_i}$, where $X := (x_1, \ldots, x_m) \in \mathbb{R}^{dm}$ is a vector. We denote $\text{Tr}(A)$ as the trace operator

of the matrix $A$ and $A^{1/2}$ is the square root of matrix $A$ i.e., $A^{1/2} = B$ such that $BB = A$. Given a probability measure $\mu$, we denote $F_\mu$ is the cumulative distribution function (CDF) of $\mu$.

## 2 BACKGROUND

We now review some essential materials for the paper.

**Wasserstein distance.** Given $p \geq 1$, two probability measures $\mu \in \mathcal{P}_p(\mathbb{R}^d)$ and $\nu \in \mathcal{P}_p(\mathbb{R}^d)$, the Wasserstein distance (Villani, 2008a; Peyré & Cuturi, 2019) between $\mu$ and $\nu$ is : $W_p^p(\mu, \nu) = \inf_{\pi \in \Pi(\mu,\nu)} \int_{\mathbb{R}^d \times \mathbb{R}^d} \|x - y\|_p^p d\pi(x, y)$, where $\Pi(\mu, \nu)$ is set of all couplings that have marginals are $\mu$ and $\nu$ respectively. The computational complexity and memory complexity of Wasserstein distance are $\mathcal{O}(n^3 \log n)$ and $\mathcal{O}(n^2)$ in turn when $\mu$ and $\nu$ have at most $n$ supports. However, there are some special cases where the Wasserstein distance can be computed efficiently.

**Special Cases.** When $p = 2$, we have Gaussian distributions $\mu := \mathcal{N}(\mathbf{m}_1, \Sigma_1)$ and $\nu := \mathcal{N}(\mathbf{m}_2, \Sigma_2)$, the Wasserstein distance between $\mu$ and $\nu$ has the following closed-form:

$$W_2^2(\mu, \nu) = \|\mathbf{m}_1 - \mathbf{m}_2\|_2^2 + \text{Tr}(\Sigma_1 + \Sigma_2 - 2(\Sigma_1^{1/2}\Sigma_2\Sigma_1^{1/2})^{1/2}). \tag{1}$$

When $d = 1$, the Wasserstein distance between $\mu \in \mathcal{P}_p(\mathbb{R})$ and $\nu \in \mathcal{P}_p(\mathbb{R})$ can also be computed with a closed form: $W_p^p(\mu, \nu) = \int_0^1 |F_\mu^{-1}(z) - F_\nu^{-1}(z)|^p dz$. When $\mu$ and $\nu$ are one-dimension discrete probability measures that have a most $n$ supports, the computational complexity and memory complexity, in this case, are only $\mathcal{O}(n \log n)$ and $\mathcal{O}(n)$.

**Sliced Wasserstein distance.** Using random projections, the sliced Wasserstein (SW) distance can exploit the closed-form benefit of Wasserstein distance in one dimension. The definition of sliced Wasserstein distance (Bonneel et al., 2015) between two probability measures $\mu \in \mathcal{P}_p(\mathbb{R}^d)$ and $\nu \in \mathcal{P}_p(\mathbb{R}^d)$ is:

$$SW_p^p(\mu, \nu) = \mathbb{E}_{\theta \sim \mathcal{U}(\mathbb{S}^{d-1})}[W_p^p(\theta\sharp\mu, \theta\sharp\nu)], \tag{2}$$

where $\mathcal{U}(\mathbb{S}^{d-1})$ is the uniform distribution over the unit-hyper sphere.

**Monte Carlo estimation.** The expectation in the SW is often intractable, hence, Monte Carlo samples are often used to estimate the expectation:

$$\widehat{SW}_p^p(\mu, \nu; L) = \frac{1}{L}\sum_{l=1}^{L} W_p^p(\theta_l\sharp\mu, \theta_l\sharp\nu), \tag{3}$$

where projecting directions $\theta_1, \ldots, \theta_L$ are drawn i.i.d from $\mathcal{U}(\mathbb{S}^{d-1})$. When $\mu$ and $\nu$ are empirical measures that have at most $n$ supports in $d$ dimension, the time complexity of SW is $\mathcal{O}(Ln \log n + Ldn)$. Here, $Ln \log n$ is for sorting $L$ sets of projected supports and $Ldn$ is for projecting supports to $L$ sets of scalars. Similarly, the space complexity for storing the projecting directions and the projected supports of SW is $\mathcal{O}(Ld + Ln)$. We refer to Algorithm 1 in Appendix D for the detailed algorithm for the SW.

**Variance and Error.** By using some simple transformations, we can derive the variance of the Monte Carlo approximation of the SW distance as: $\text{Var}[\widehat{SW}_p^p(\mu, \nu; L)] = \frac{1}{L}\text{Var}[W_p^p(\theta\sharp\mu, \theta\sharp\nu)]$. We also have the error of the Monte Carlo approximation (Nadjahi et al., 2020b) is:

$$\mathbb{E}\left[\left|\widehat{SW}_p^p(\mu, \nu; L) - SW_p^p(\mu, \nu)\right|\right] \leq \frac{1}{\sqrt{L}}\text{Var}\left[W_p^p(\theta\sharp\mu, \theta\sharp\nu)\right]^{1/2}. \tag{4}$$

Here, can see that $\text{Var}[W_p^p(\theta\sharp\mu, \theta\sharp\nu)]$ plays an important role in controlling the approximation error. Therefore, a natural question arises: "Can we construct a function $Z(\theta; \mu, \nu)$ such that $\mathbb{E}[Z(\theta; \mu, \nu)] = SW_p^p(\mu, \nu)$ while $\text{Var}[Z(\theta; \mu, \nu)] \leq \text{Var}[W_p^p(\theta\sharp\mu, \theta\sharp\nu)]$?". If we can design such $Z(\theta; \mu, \nu)$, we will have $\mathbb{E}\left[\left|\frac{1}{L}\sum_{l=1}^{L} Z(\theta_l; \mu, \nu) - SW_p^p(\mu, \nu)\right|\right] \leq \frac{1}{\sqrt{L}}\text{Var}[Z(\theta; \mu, \nu)]^{1/2} \leq \frac{1}{\sqrt{L}}\text{Var}[W_p^p(\theta\sharp\mu, \theta\sharp\nu)]^{1/2}$ which implies that the estimator $\frac{1}{L}\sum_{l=1}^{L} Z(\theta_l; \mu, \nu)$ is better than $\widehat{SW}_p^p(\mu, \nu; L)$.

## 3 CONTROL VARIATE SLICED WASSERSTEIN ESTIMATORS

We first adapt notations from the control variates literature to the SW case in Section 3.1. After that, we discuss how to construct control variates and their computational properties in Section 3.2.

### 3.1 CONTROL VARIATE FOR SLICED WASSERSTEIN DISTANCE

**Short Notations.** We are approximating the SW which is an expectation with respect to the random variable $\theta$. For convenience, let denote $W_p^p(\theta\sharp\mu, \theta\sharp\nu)$ as $W(\theta; \mu, \nu)$, we have:

$$\mathrm{SW}_p^p(\mu, \nu) = \mathbb{E}[W(\theta; \mu, \nu)], \quad \text{and} \quad \widehat{\mathrm{SW}}_p^p(\mu, \nu; L) = \frac{1}{L}\sum_{l=1}^{L} W(\theta_l; \mu, \nu), \tag{5}$$

where $\theta_1, \ldots, \theta_L \overset{i.i.d}{\sim} \sigma_0(\theta) := \mathcal{U}(\mathbb{S}^{d-1})$.

**Control Variate.** A *control variate* (Owen, 2013) is a random variable $C(\theta)$ such that its expectation is tractable i.e., $\mathbb{E}[C(\theta)] = B$. From the control variate, we consider the following variable:

$$W(\theta; \mu, \nu) - \gamma(C(\theta) - B) \tag{6}$$

where $\gamma \in \mathbb{R}$. Therefore, it is easy to check that $\mathbb{E}[W(\theta; \mu, \nu) - \gamma(C(\theta) - B)] = \mathbb{E}[W(\theta; \mu, \nu)] = \mathrm{SW}_p^p(\mu, \nu)$ since $\mathbb{E}[C(\theta)] = B$. Therefore, the Monte Carlo estimation of $\mathbb{E}[W(\theta; \mu, \nu) - \gamma(C(\theta) - B)]$, i.e., $\frac{1}{L}\sum_{l=1}^{L}(W(\theta_k; \mu, \nu) - \gamma(C(\theta_l) - B))$, is an unbiased estimation of $\mathbb{E}[W(\theta; \mu, \nu)]$.

Now, we consider the variance of the variable $W(\theta; \mu, \nu) - \gamma(C(\theta) - B)$.

$$\mathrm{Var}[W(\theta; \mu, \nu) - \gamma(C(\theta) - B)] = \mathrm{Var}[W(\theta; \mu, \nu)] - 2\gamma\mathrm{Cov}[W(\theta; \mu, \nu), C(\theta)] + \gamma^2\mathrm{Var}[C(\theta)].$$

The above variance attains its minimum, with respect to $\gamma$, for $\gamma^\star = \frac{\mathrm{Cov}[W(\theta;\mu,\nu),C(\theta)]}{\mathrm{Var}[C(\theta)]}$ Using the optimal $\gamma^\star$, we the minimum variance of is:

$$\mathrm{Var}[W(\theta; \mu, \nu)]\left(1 - \frac{\mathrm{Cov}[W(\theta;\mu,\nu),C(\theta)]^2}{\mathrm{Var}[W(\theta;\mu,\nu)]\mathrm{Var}[C(\theta)]}\right). \tag{7}$$

Therefore, the variance of $W(\theta; \mu, \nu) - \gamma^\star(C(\theta) - B)$ with is **lower** than $W(\theta; \mu, \nu)$ if the control variate $C(\theta)$ has a correlation with $W(\theta; \mu, \nu)$.

**Definition 1.** *Given a control variate $C(\theta)$, the corresponding controlled projected one-dimensional Wasserstein distance is:*

$$Z(\theta; \mu, \nu, C(\theta)) = W(\theta; \mu, \nu) - \frac{Cov[W(\theta;\mu,\nu),C(\theta)]}{Var[C(\theta)]}(C(\theta) - B). \tag{8}$$

In practice, computing $\gamma^\star$ might be intractable, hence, we can estimate $\gamma^\star$ using Monte Carlo samples.

**Control Variate Estimator of the SW distance.** Now, we can form the new estimation of $\mathrm{SW}_p^p(\mu, \nu)$.

**Definition 2.** *Given a control variate $C(\theta)$ with $\mathbb{E}[C(\theta)] = B$, the number of projections $L \geq 1$, the Control Variate Sliced Wasserstein estimator is:*

$$\widehat{\mathit{CV\text{-}SW}}_p^p(\mu, \nu; L, C(\theta)) = \widehat{SW}_p^p(\mu, \nu; L) - \widehat{\gamma^\star}_L\frac{1}{L}\sum_{l=1}^{L}(C(\theta_l) - B), \tag{9}$$

*where $\theta_1, \ldots, \theta_L \sim \sigma_0(\theta)$, $\widehat{SW}_p^p(\mu, \nu; L) = \frac{1}{L}\sum_{l=1}^{L} W(\theta_l; \mu, \nu)$, and the estimated optimal coefficient $\widehat{\gamma^\star}_L = \frac{\widehat{Cov}[W(\theta;\mu,\nu),C(\theta);L]}{\widehat{Var}[C(\theta);L]}$ with $\widehat{Cov}[W(\theta;\mu,\nu),C(\theta);L] = \frac{1}{L}\sum_{l=1}^{L}(W(\theta_l;\mu,\nu) - \widehat{SW}_p^p(\mu,\nu;L))(C(\theta_l) - B)$, $\widehat{Var}[C(\theta);L] = \frac{1}{L}\sum_{l=1}^{L}(C(\theta_l) - B)^2$.*

**Remark 1.** *In Definition 2, the optimal coefficient $\widehat{\gamma^\star}_L$ is estimated by reusing Monte Carlo samples $\theta_1, \ldots, \theta_L$. It is possible to estimate $\widehat{\gamma^\star}_L$ using new samples $\theta'_1, \ldots, \theta'_L$ to make it independent to $\widehat{SW}_p^p(\mu, \nu; L)$. Nevertheless, this resampling approach costs doubling computational complexity. Therefore, the estimation in Definition 2 is preferable in practice (Owen, 2013). It is worth noting that the estimation is only asymptotic unbiased, however, it is negligible since the unbiasedness is always lost after taking the p-root for computing the value of SW with finite $L$.*

In this section, we have not yet specified $C(\theta)$. In the next session, we will focus on constructing a computationally efficient $C(\theta)$ that is also effective by conditioning it on two probability measures $\mu$ and $\nu$. Specifically, we define $C(\theta) := C(\theta; \mu, \nu)$, which involves conditioning on these two probability measures $\mu$ and $\nu$.

## 3.2 Constructing Control Variates

We recall that we are interested in the random variable $W(\theta; \mu, \nu) = W_p^p(\theta\sharp\mu, \theta\sharp\nu)$. It is desirable to have a control variate $C(\theta)$ such that $C(\theta) \approx W(\theta; \mu, \nu)$ (Owen, 2013). Therefore, it is natural to also construct $C(\theta) := C(\theta; \mu, \nu)$ based on two measures $\mu, \nu$. Moreover, since we know the closed-form solution of Wasserstein distance between two Gaussians, control variates in the form $C(\theta; \mu, \nu) = W_2^2(\mathcal{N}(m_1(\theta; \mu), \sigma_1^2(\theta; \mu)), \mathcal{N}(m_2(\theta; \nu), \sigma_2^2(\theta; \nu)))$ could be tractable.

**Gaussian Approximation.** The question of constructing the control variate i.e., specifying $\mathcal{N}(m_1(\theta; \mu), \sigma_1^2(\theta; \mu))$ and $\mathcal{N}(m_2(\theta; \nu), \sigma_2^2(\theta; \nu))$ now requires finding the Gaussian approximation of two projected measures $\theta\sharp\mu$ and $\theta\sharp\nu$. When two measures are discrete, namely, $\mu = \sum_{i=1}^n \alpha_i \delta_{x_i}$ ($\sum_{i=1}^n \alpha_i = 1$) and $\nu = \sum_{i=1}^m \beta_i \delta_{y_i}$ ($\sum_{i=1}^m \beta_i = 1$), we perform the following optimization:

$$m_1(\theta; \mu), \sigma_1^2(\theta; \mu) = \operatorname{argmax}_{m_1, \sigma_1^2} \left[ \sum_{i=1}^n \alpha_i \log \left( \frac{1}{\sqrt{2\pi\sigma_1^2}} \exp\left( -\frac{1}{2\sigma_1^2} (\theta^\top x_i - m_1)^2 \right) \right) \right], \quad (10)$$

$$m_2(\theta; \nu), \sigma_2^2(\theta; \nu) = \operatorname{argmax}_{m_2, \sigma_2^2} \left[ \sum_{i=1}^m \beta_i \log \left( \frac{1}{\sqrt{2\pi\sigma_2^2}} \exp\left( -\frac{1}{2\sigma_2^2} (\theta^\top y_i - m_2)^2 \right) \right) \right], \quad (11)$$

**Proposition 1.** *Let $\mu$ and $\mu$ be two discrete probability measures i.e., $\mu = \sum_{i=1}^n \alpha_i \delta_{x_i}$ ($\sum_{i=1}^n \alpha_i = 1$) and $\nu = \sum_{i=1}^m \beta_i \delta_{y_i}$ ($\sum_{i=1}^m \beta_i = 1$), we have: $m_1(\theta; \mu) = \sum_{i=1}^n \alpha_i \theta^\top x_i$, $\sigma_1^2(\theta; \mu) = \sum_{i=1}^n \alpha_i \left( \theta^\top x_i - m_1(\theta; \mu) \right)^2$, $m_2(\theta; \nu) = \sum_{i=1}^m \beta_i \theta^\top y_i$, $\sigma_2^2(\theta; \nu) = \sum_{i=1}^m \beta_i \left( \theta^\top y_i - m_2(\theta; \nu) \right)^2$, are solution of the problems in equation 10- 11.*

We refer to Appendix C.1 for the detailed proof of Proposition 1. It is worth noting that the closed-form in Proposition 1 can also be seen as the solution from using moment matching. When $\mu$ and $\nu$ are continuous, we can solve the optimization by using stochastic gradient descent by sampling from $\mu$ and $\nu$ respectively. In addition, we can use corresponding empirical measures of $\mu$ and $\nu$ i.e., $\mu_n = \frac{1}{n} \sum_{i=1}^n \delta_{X_i}$ and $\nu_n = \frac{1}{n} \sum_{i=1}^n \delta_{Y_i}$ as proxies (where $X_1, \ldots, X_n \overset{i.i.d}{\sim} \mu$ and $Y_1, \ldots, Y_n \overset{i.i.d}{\sim} \nu$). Beyond optimization, we can also utilize the Laplace approximation to obtain two approximated Gaussians for $\theta\sharp\mu$ and $\theta\sharp\nu$ when the push-forward densities are tractable. We refer the reader to Appendix A for more detail.

**Constructing Control Variates.** From the closed-form of the Wasserstein-2 distance between two Gaussians in equation 1, we have: $W_2^2(\mathcal{N}(m_1(\theta; \mu), \sigma_1^2(\theta; \mu)), \mathcal{N}(m_2(\theta; \nu), \sigma_2^2(\theta; \nu))) = (m_1(\theta; \mu) - m_2(\theta; \nu))^2 + \sigma_1(\theta; \mu)^2 + \sigma_2(\theta; \nu)^2 - 2\sigma_1(\theta; \mu)\sigma_2(\theta; \nu)$. To calculate $\mathbb{E}[W_2^2(\mathcal{N}(m_1(\theta; \mu), \sigma_1^2(\theta; \mu)), \mathcal{N}(m_2(\theta; \nu), \sigma_2^2(\theta; \nu)))]$ as the requirement of a control variate, we could calculate the expectation of each term i.e., $\mathbb{E}[(m_1(\theta; \mu) - m_2(\theta; \nu))^2]$, $\mathbb{E}\left[\sigma_1^2(\theta; \mu)\right]$, $\mathbb{E}\left[\sigma_2^2(\theta; \nu)\right]$, and $\mathbb{E}[\sigma_1(\theta; \mu)\sigma_2(\theta; \nu)]$.

**Proposition 2.** *Given two discrete measures $\mu$ and $\nu$, $m_1(\theta; \mu)$, $m_2(\theta; \nu)$, $\sigma_1^2(\theta; \mu)$, and $\sigma_1^2(\theta; \mu)$ as in Proposition 1, we obtain:*

$$\mathbb{E}[(m_1(\theta; \mu) - m_2(\theta; \nu))^2] = \frac{1}{d} \left\| \sum_{i=1}^n \alpha_i x_i - \sum_{i=1}^m \beta_i y_i \right\|^2, \quad (12)$$

$$\mathbb{E}\left[\sigma_1^2(\theta; \mu)\right] = \frac{1}{d} \sum_{i=1}^n \alpha_i \left\| x_i - \sum_{i'=1}^n \alpha_{i'} x_{i'} \right\|^2, \quad \mathbb{E}\left[\sigma_2^2(\theta; \nu)\right] = \frac{1}{d} \sum_{i=1}^m \beta_i \left\| y_i - \sum_{i'=1}^m \beta_{i'} y_{i'} \right\|^2 \quad (13)$$

The proof of Proposition 2 is given in Appendix C.2.

Unfortunately, we cannot compute the expectation for the last term $\mathbb{E}[\sigma_1(\theta; \mu)\sigma_2(\theta; \nu)]$. However, we could construct control variates using lower-bound and upper-bound of $W_2^2(\mathcal{N}(m_1(\theta; \mu), \sigma_1^2(\theta; \mu)), \mathcal{N}(m_2(\theta; \nu), \sigma_2^2(\theta; \nu)))$.

**Definition 3.** *Given two Gaussian approximations of $\theta\sharp\mu$ and $\theta\sharp\nu$ i.e., $\mathcal{N}(m_1(\theta; \mu), \sigma_1^2(\theta; \mu))$ and $\mathcal{N}(m_2(\theta; \nu), \sigma_2^2(\theta; \nu))$, we define the following two control variates:*

$$C_{low}(\theta; \mu, \nu) = (m_1(\theta; \mu) - m_2(\theta; \nu))^2 \quad (14)$$

$$C_{up}(\theta; \mu, \nu) = (m_1(\theta; \mu) - m_2(\theta; \nu))^2 + \sigma_1^2(\theta; \mu) + \sigma_2^2(\theta; \nu). \quad (15)$$

We now discuss some properties of the proposed control variates.

**Proposition 3.** *We have the following relationship:*

$$C_{low}(\theta; \mu, \nu) \le W_2^2(\mathcal{N}(m_1(\theta; \mu), \sigma_1^2(\theta; \mu)), \mathcal{N}(m_2(\theta; \nu), \sigma_2^2(\theta; \nu))) \le C_{up}(\theta; \mu, \nu). \quad (16)$$

The proof of Proposition 3 follows directly from the construction of the control variates. For completeness, we provide the proof in Appendix C.3.

**Proposition 4.** *Let $\mu = \sum_{i=1}^n \alpha_i \delta_{x_i}$ and $\nu = \sum_{i=1}^m \beta_i \delta_{y_i}$, using the control variate $C_{low}(\theta; \mu, \nu)$ is equivalent to using the following control variate:*

$$W_2^2(\theta \sharp \mathcal{N}(\mathbf{m_1}(\mu), \sigma_1^2(\mu)\mathbf{I}), \theta \sharp \mathcal{N}(\mathbf{m_2}(\nu), \sigma_2^2(\nu)\mathbf{I})), \quad (17)$$

*where $\mathbf{m_1}(\mu), \sigma_1^2(\mu) = argmax_{\mathbf{m_1}, \sigma_1^2} \sum_{i=1}^n \alpha_i \log \left( \frac{1}{\sqrt{(2\pi)^d |\sigma_1^2 \mathbf{I}|}} e^{\left(-\frac{1}{2}(x_i - \mathbf{m_1})^\top |\sigma_1^2 \mathbf{I}|^{-1}(x_i - \mathbf{m_1})\right)} \right)$*

*and $\mathbf{m_2}(\nu), \sigma_2^2(\nu) = argmax_{\mathbf{m_2}, \sigma_2^2} \sum_{i=1}^m \beta_i \log \left( \frac{1}{\sqrt{(2\pi)^d |\sigma_2^2 \mathbf{I}|}} e^{\left(-\frac{1}{2}(y_i - \mathbf{m_2})^\top |\sigma_2^2 \mathbf{I}|^{-1}(y_i - \mathbf{m_2})\right)} \right)$.*

The proof for The Proposition 4 is given in Appendix C.4. The proposition means that using the control variate $C_{low}(\theta; \mu, \nu)$ is equivalent to using the Wasserstein-2 distance between two projected *location-scale multivariate* Gaussians with these location-scale multivariate Gaussians are the approximation of the two original probability measures $\mu$ and $\nu$ on the original space.

**Control Variate Estimators of the SW distance.** From two defined control variates in Definition 3, we define the corresponding controlled projected one-dimensional Wasserstein distances and control variate sliced Wasserstein estimators.

**Definition 4.** *Given two control variates $C_{low}(\theta; \mu, \nu), C_{up}(\theta; \mu, \nu)$ in Definition 3, the corresponding controlled projected one-dimensional Wasserstein distances are defined as:*

$$Z_{low}(\theta; \mu, \nu) = Z(\theta; \mu, \nu, C_{low}(\theta; \mu, \nu)), \quad Z_{up}(\theta; \mu, \nu) = Z(\theta; \mu, \nu, C_{up}(\theta; \mu, \nu)), \quad (18)$$

*where $Z(\theta; \mu, \nu, C(\theta))$ is defined in Definition 1.*

**Definition 5.** *Given $Z_{low}(\theta, \mu, \nu)$, $Z_{up}(\theta, \mu, \nu)$, $\widehat{CV\text{-}SW}_p^p(\mu, \nu; L, C(\theta))$ as defined in Definition 4 and Definition 4, the lowerbound and upperbound control variate sliced Wasserstein estimators are:*

$$\widehat{LCV\text{-}SW}_p^p(\mu, \nu; L) = \frac{1}{L} \sum_{i=1}^L Z_{low}(\theta_l, \mu, \nu) = \widehat{CV\text{-}SW}_p^p(\mu, \nu; L, C_{low}(\theta; \mu, \nu)), \quad (19)$$

$$\widehat{UCV\text{-}SW}_p^p(\mu, \nu; L) = \frac{1}{L} \sum_{i=1}^L Z_{up}(\theta_l, \mu, \nu) = \widehat{CV\text{-}SW}_p^p(\mu, \nu; L, C_{up}(\theta; \mu, \nu)). \quad (20)$$

We refer to Algorithm 2- 3 for the detailed algorithms for the control variate estimators in Appendix D.

**Computational Complexities.** When dealing with two discrete probability measures $\mu$ and $\nu$ that have at most $n$ supports, projecting the supports of the two measures to $L$ sets of projected supports costs $\mathcal{O}(Ldn)$ in time complexity. After that, fitting one-dimensional Gaussians also costs $\mathcal{O}(Ldn)$. From Proposition 2 and Definition 3, computing the control variate $C_{low}(\theta; \mu, \nu)$ and its expected value $B_{low}(\mu, \nu)$ costs $\mathcal{O}(dn)$. Similarly, computing the control variate $C_{up}(\theta; \mu, \nu)$ and its expected value $B_{up}$ also costs $\mathcal{O}(dn)$. Overall, the time complexity of $\widehat{LCV\text{-}SW}_p^p(\mu, \nu; L)$ and $\widehat{UCV\text{-}SW}_p^p(\mu, \nu; L)$ is the same as the conventional SW, which is $\mathcal{O}(Ln \log n + Ldn)$. Similarly, their space complexities are also $\mathcal{O}(Ld + Ln)$ as the conventional estimator.

**Gaussian Approximation of the sliced Wasserstein distance.** In a recent work (Nadjahi et al., 2021), the closed-form of Wasserstein between Gaussian distributions is utilized to approximate the value of the SW distance. The key idea is based on the concentration of random projections in high-dimension, i.e., they are approximately Gaussian. However, the resulting deterministic approximation is not very accurate since it is only based on the first two moments of two original measures. As a result, the proposed approximation can only work well when two measures have weak dimensional dependence. In contrast, we use the closed-form of Wasserstein between Gaussian distributions to design control variates, which can reduce the variance of the Monte Carlo estimator.

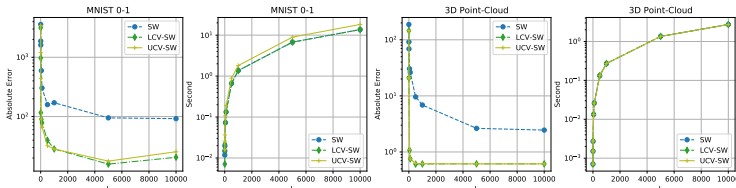

Figure 1: The empirical errors of the conventional estimator (SW) and the control variate estimators (LCV-SW, UCV-SW) when comparing empirical distributions over MNIST images and point-clouds.

Table 1: Summary of Wasserstein-2 (Flamary et al., 2021) (multiplied by $10^4$), computational time in second (s) to reach step 8000 of different sliced Wasserstein estimators in gradient flows from three different runs.

| Distances | Step 3000 (W$_2$ ↓) | Step 4000 (W$_2$ ↓) | Step 5000 (W$_2$ ↓) | Step 6000(W$_2$ ↓) | Step 8000 (W$_2$ ↓) | Time (s ↓) |
|---|---|---|---|---|---|---|
| SW L=10 | $305.5196 \pm 0.4921$ | $137.8767 \pm 0.3631$ | $36.2454 \pm 0.1387$ | $0.1164 \pm 0.0022$ | $2.1e-5 \pm 1.0e-5$ | $\mathbf{26.13 \pm 0.04}$ |
| LCV-SW L=10 | $\mathbf{303.1001 \pm 0.1787}$ | $\mathbf{136.0129 \pm 0.0923}$ | $35.0929 \pm 0.1459$ | $0.0538 \pm 0.0047$ | $1.5e-5 \pm 0.2e-5$ | $27.22 \pm 0.24$ |
| UCV-SW L=10 | $303.1017 \pm 0.1783$ | $136.0139 \pm 0.0921$ | $\mathbf{35.0916 \pm 0.1444}$ | $\mathbf{0.0535 \pm 0.0065}$ | $\mathbf{1.4e-5 \pm 0.2e-5}$ | $29.48 \pm 0.23$ |
| SW L=100 | $300.7326 \pm 0.2375$ | $134.1498 \pm 0.3146$ | $33.9253 \pm 0.1349$ | $0.0183 \pm 0.0011$ | $1.6e-5 \pm 0.2e-5$ | $220.20 \pm 0.21$ |
| LCV-SW L=100 | $\mathbf{300.3924 \pm 0.0053}$ | $133.6243 \pm 0.0065$ | $33.5102 \pm 0.0031$ | $\mathbf{0.0134 \pm 8.7e-5}$ | $\mathbf{1.4e-5 \pm 0.1e-5}$ | $221.75 \pm 0.35$ |
| UCV-SW L=100 | $300.3927 \pm 0.0050$ | $\mathbf{133.6242 \pm 0.0065}$ | $\mathbf{33.5088 \pm 0.0028}$ | $0.0136 \pm 0.0003$ | $1.4e-5 \pm 0.2e-5$ | $233.71 \pm 0.06$ |

Our estimators are still stochastic, converge to the true value of the SW when increasing the number of projections, and work well with probability measures that have arbitrary structures and dimensions.

**Beyond uniform slicing distribution and related works.** In the SW distance, the projecting direction follows the uniform distribution over the unit-hypersphere $\sigma_0(\theta) = \mathcal{U}(\mathbb{S}^{d-1})$. In some cases, changing the slicing distribution might benefit downstream applications. For example, the distributional sliced Wasserstein distance (Nguyen et al., 2021) can be written as $\mathrm{DSW}_p^p(\mu, \nu) = \mathbb{E}_{\theta \sim \sigma(\theta)} \left[ \mathrm{W}_p^p(\theta \sharp \mu, \theta \sharp \nu) \right]$, where $\sigma(\theta)$ is a distribution over the unit hypersphere. We can directly apply the proposed control variates to the DSW as long as we can calculate $\mathbb{E}_{\theta \sim \sigma(\theta)}[\theta \theta^\top]$. If this is not the case, we can still use the proposed control variates by approximating the DSW via importance sampling. In particular, we can rewrite $\mathrm{DSW}_p^p(\mu, \nu) = \mathbb{E}_{\theta \sim \sigma_0(\theta)} \left[ \mathrm{W}_p^p(\theta \sharp \mu, \theta \sharp \nu) \frac{\sigma(\theta)}{\sigma_0(\theta)} \right]$, then use control variates. We discuss other related works including how we can apply the control variates to other variants of the SW distance in Appendix E.

**Gradient estimators.** In some cases, we might be interested in obtaining the gradient $\nabla_\phi \mathrm{SW}_p^p(\mu, \nu_\phi) = \nabla_\phi \mathbb{E}[\mathrm{W}_p^p(\theta \sharp \mu, \theta \sharp \nu_\phi)]$ e.g., statistical inference. Since the expectation in the SW does not depend on $\phi$, we can use the Leibniz rule to exchange differentiation and expectation, namely, $\nabla_\phi \mathbb{E}[\mathrm{W}_p^p(\theta \sharp \mu, \theta \sharp \nu_\phi)] = \mathbb{E}[\nabla_\phi \mathrm{W}_p^p(\theta \sharp \mu, \theta \sharp \nu_\phi)]$. After that, we can use $\nabla_\phi C(\theta; \mu, \nu_\phi)$ ($\nabla_\phi C_{low}(\theta; \mu, \nu_\phi)$ or $\nabla_\phi C_{up}(\theta; \mu, \nu_\phi)$) as the control variate. However, estimating the optimal $\gamma_i^\star = \frac{\mathrm{Cov}[\nabla_{\phi_i} \mathrm{W}_p^p(\theta \sharp \mu, \theta \sharp \nu_\phi), \nabla_{\phi_i} C(\theta; \mu, \nu_\phi)]}{\mathrm{Var}[\nabla_{\phi_i} C(\theta; \mu, \nu_\phi)]}$ (for $i = 1, \ldots, d'$ with $d'$ is the number of dimensions of parameters $\phi$) is computationally expensive and depends significantly on the specific settings of $\nu_\phi$. Therefore, we utilize a more computationally efficient estimator of the gradient i.e., $\mathbb{E}[\nabla_\phi Z(\theta; \mu, \nu_\phi)]$ for $Z(\theta; \mu, \nu_\phi)$ is $Z_{low}(\theta; \mu, \nu_\phi)$ or $Z_{up}(\theta; \mu, \nu_\phi)$ in Definition 4. After that, Monte Carlo samples are used to approximate expectations to obtain a stochastic gradient.

## 4 EXPERIMENTS

Our experiments aim to show that using control variates can benefit applications of the sliced Wasserstein distance. In particular, we show that the proposed control variate estimators have lower variance in practice when comparing probability measures over images and point-clouds in Section 4.1. After that, we show that using the proposed control variate estimators can help to drive a gradient flow to converge faster to a target probability measure from a source probability measure while retaining approximately the same computation in Section 4.2. Finally, we show the proposed control variate estimators can also improve training deep generative models on standard benchmark images dataset such as CIFAR10, and CelebA in Section 4.3. Due to the space constraint, additional experiments and detailed experimental settings are deferred to Appendix F.

### 4.1 COMPARING EMPIRICAL PROBABILITY MEASURES OVER IMAGES AND POINT-CLOUDS

**Settings.** We aim to first test how well the proposed control variates can reduce the variance in practice. To do this, we use conventional Monte Carlo estimation for the $SW_2^2$ with a very

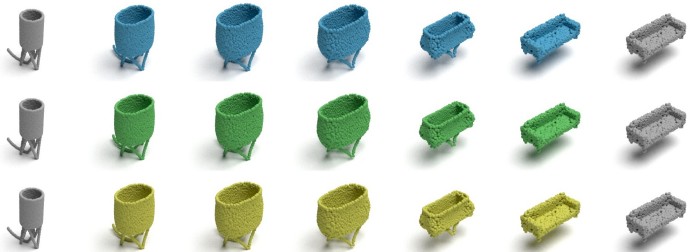

Figure 2: Point-cloud gradient flows for $L = 10$ from SW, LCV-SW, and UCV-SW respectively.

large $L$ value of 100000 and treat it as the population value. Next, we vary $L$ in the set $\{2, 5, 10, 50, 100, 500, 1000, 5000, 7000, 10000\}$ and compute the corresponding estimates of the SW with both the conventional estimator and control variate estimators. Finally, we calculate the absolute difference between the estimators and the estimated population as the estimated error. We apply this process to compare the empirical distributions over images of digit 0 with the empirical distribution images of digit 1 in MNIST (LeCun et al., 1998) (784 dimensions, with about 6000 supports). Similarly, we compare two empirical distributions over two randomly chosen point-clouds in the ShapeNet Core-55 dataset (Chang et al., 2015) (3 dimensions, 2048 supports). We also estimate the variance of the estimators via Monte Carlo samples with $L = 100000$ in Table 3 in Appendix F.1.

**Results.** We show the estimated errors and the computational time which are averaged from 5 different runs in Figure 1. From the figure, we observe that control variate estimators reduce considerably the error in both cases. On MNIST, the lower bound control variate estimator (LCV-SW) gives a lower error than the upper bound control variate estimator (UCV-SW) while being slightly faster. The computational time of the LCV-SW has nearly identical computational time as the conventional estimator (SW). In the lower dimension i.e., the point-cloud case, two control variates give the same quality of reducing error and the same computational time. In this case, the computational time of both two control variates estimators are the same as the conventional estimator. From Table 3 in Appendix F.1, we observe that the variance of the control variate estimators is lower than the conventional estimator considerably, especially the LCV-SW. We refer to Appendix F.1 for experiments on different pairs of digits and point-clouds which show the same phenomenon.

### 4.2 POINT CLOUD GRADIENT FLOWS

**Settings.** We model a distribution $\mu(t)$ flowing with time $t$ along the sliced Wasserstein gradient flow $\mu(t) \to SW_p(\mu(t), \nu)$, which drives it towards a target distribution $\nu$ (Santambrogio, 2015). Here, we set $\nu = \frac{1}{n} \sum_{i=1}^{n} \delta_{Y_i}$ as a fixed empirical target distribution and the model distribution $\mu(t) = \frac{1}{n} \sum_{i=1}^{n} \delta_{X_i(t)}$, where the time-varying point cloud $X(t) = (X_i(t))_{i=1}^{n} \in (\mathbb{R}^d)^n$. Starting at time $t = 0$, we integrate the ordinary differential equation $\dot{X}(t) = -n \nabla_{X(t)} \left[ SW_p \left( \frac{1}{n} \sum_{i=1}^{n} \delta_{X_i(t)}, \nu \right) \right]$ for each iteration. We choose $\mu(0)$ and $\nu$ are two random point-cloud shapes in ShapeNet Core-55 dataset (Chang et al., 2015) which were used in Section 4.1. After that, we use different estimators of the SW i.e., $\widehat{SW}_p$, $\widehat{LCV\text{-}SW}_p$, and $\widehat{UCV\text{-}SW}_p$, as the replacement of the for $SW_p$ to estimate the gradient including the proposed control variate estimators and the conventional estimator. Finally, we use $p = 2$, and the Euler scheme with 8000 iterations and step size 0.01 to solve the flows.

**Results.** We use the Wasserstein-2 distance as a neutral metric to evaluate how close the model distribution $\mu(t)$ is to the target distribution $\nu$. We show the results for the conventional Monte Carlo estimator (denoted as SW), the control variate $C_{low}$ estimator (denoted as LCV-SW), and the control variate $C_{up}$ estimator (denoted as UCV-SW), with the number of projections $L = 10, 100$ in Table 1. In the table, we also report the time in seconds that estimators need to reach the last step. In addition, we visually the flows for $L = 100$ in Figure 2. Moreover, we observe the same phenomenon on a different pair of point-clouds in Table 4 and in Figure 6 in Appendix F.2.

### 4.3 DEEP GENERATIVE MODELING

**Settings.** We conduct deep generative modeling on CIFAR10 (with image size 32x32) (Krizhevsky et al., 2009), *non-croppeed* CelebA (with image size 64x64) (Liu et al., 2015) with the SNGAN

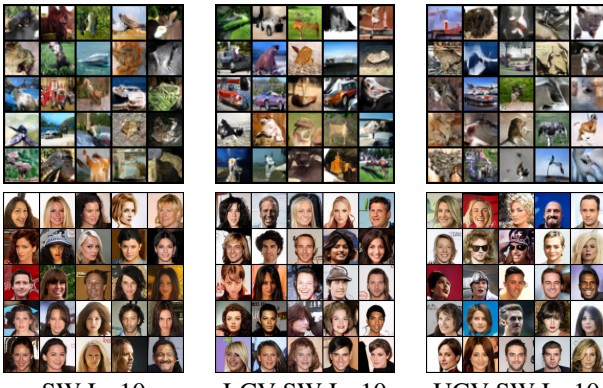

SW L=10        LCV-SW L=10      UCV-SW L=10

Figure 3: Random generated images of distances on CIFAR10 and CelebA.

Table 2: Summary of FID and IS scores from three different runs on CIFAR10 (32x32), and CelebA (64x64).

| Method | CIFAR10 (32x32) | | CelebA (64x64) | Method | CIFAR10 (32x32) | | CelebA (64x64) |
|---|---|---|---|---|---|---|---|
| | FID ($\downarrow$) | IS ($\uparrow$) | FID ($\downarrow$) | | FID ($\downarrow$) | IS ($\uparrow$) | FID ($\downarrow$) |
| SW L=10 | $21.72 \pm 0.37$ | $7.54 \pm 0.06$ | $11.14 \pm 1.12$ | SW L=1000 | $14.07 \pm 0.97$ | $8.07 \pm 0.06$ | $10.05 \pm 0.40$ |
| LCV-SW L=10 | $\mathbf{18.67 \pm 0.39}$ | $\mathbf{7.74 \pm 0.04}$ | $\mathbf{10.17 \pm 0.73}$ | LCV-SW L=1000 | $13.58 \pm 0.12$ | $\mathbf{8.23 \pm 0.02}$ | $9.78 \pm 0.47$ |
| UCV-SW L=10 | $21.71 \pm 0.36$ | $7.57 \pm 0.01$ | $11.00 \pm 0.04$ | UCV-SW L=1000 | $\mathbf{13.21 \pm 0.49}$ | $8.21 \pm 0.09$ | $\mathbf{9.51 \pm 0.30}$ |

architecture (Miyato et al., 2018). For the framework, we follow the sliced Wasserstein genera­tor (Deshpande et al., 2018; Nguyen & Ho, 2023) which is described in detail in Appendix F.3. The main evaluation metrics are FID score (Heusel et al., 2017) and Inception score (IS) (Salimans et al., 2016). On CelebA, we do not report the IS since it poorly captures the perceptual quality of face images (Heusel et al., 2017). The detailed settings about architectures, hyper-parameters, and evaluation of FID and IS are provided in Appendix F.3.

**Results.** We train generators with the conventional estimator (SW), and the proposed control variates estimators (LCV-SW and UCV-SW) with the number of projections $L = 10, 1000$. We report FID scores and IS scores in Table 2. From the table, we observe that the LCV-SW gives the best generative models on both CIFAR10 and CelebA when $L = 10$. In particular, the LCV-SW can reduce about $14\%$ FID score on CIFAR10, and about $10\%$ FID score on CelebA. Also, the LCV-SW increases the IS score by about $2.6\%$. For the UCV-SW, it can enhance the performance slightly compared with the SW when $L = 10$. It is worth noting that, the computational times of estimators are approximately the same since the main computation in the application is for training neural networks. Therefore, we can see the benefit of reducing the variance by using control variates here. Increasing $L$ to $1000$, despite the gap being closer, the control variate estimators still give better scores than the conventional estimator. In greater detail, the LCV-SW can improve about $3.5\%$ FID score on CIFAR10, about $2\%$ IS score on CIFAR10, and about $2.7\%$ FID score on CelebA compared to the SW. In this setting of $L$, the UCV-SW gives the best FID scores on both CIFAR10 and CelebA. Concretely, it helps to decrease the FID score by about $6.1\%$ on CIFAR10 and about $5.4\%$ on CelebA. From the result, the UCV-SW seems to need more projections than the LCV-SW to approximate well the optimal control variate coefficient ($\gamma^\star$) since the corresponding control variate of the UCV-SW has two more random terms. For the qualitative result, we show randomly generated images from trained models in Figure 3 for $L = 10$ and in Figure 7 for $L = 1000$ in Appendix F.3. Overall, we obverse that generated images are visually consistent to the FID scores and IS scores in Table 2.

## 5 CONCLUSION

We have presented a method for reducing the variance of Monte Carlo estimation of the sliced Wasser­stein (SW) distance using control variates. Moreover, we propose two novel control variates that serve as lower and upper bounds for the Wasserstein distance between two Gaussian approximations of two measures. By using the closed-form solution of the Wasserstein distance between two Gaussians and the closed-form of Gaussian fitting of discrete measures via Kullback Leibler divergence, we demonstrate that the proposed control variates can be computed in linear time. Consequently, the control variate estimators have the same computational complexity as the conventional estimator of SW distance. On the experimental side, we demonstrate that the proposed control variate estimators have smaller variances than the conventional estimator when comparing probability measures over images and point clouds. Finally, we show that using the proposed control variate estimators can lead to improved performance of point-cloud SW gradient flows and generative models.

## ACKNOWLEDGEMENTS

We would like to thank Peter Mueller for his insightful discussion during the course of this project. NH acknowledges support from the NSF IFML 2019844 and the NSF AI Institute for Foundations of Machine Learning.

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

# Supplement to "Sliced Wasserstein Estimation with Control Variates"

We first provide the skipped proofs in Appendix C. Additionally, we present the detailed algorithms for the conventional estimator of the sliced Wasserstein distance and the control variate estimators in Appendix D. Sliced Wasserstein variants are discussed in Appendix E, and we also discuss how to use control variates in those variants in the same appendix. In Appendix F, we provide additional experiments and their detailed experimental settings. Finally, we report the computational infrastructure in Appendix G.

## A  DISCUSSION

**Laplace Approximation.** We are interested in approximating a continuous distribution $\mu$ by an Gaussian $\mathcal{N}(m, \sigma^2)$. We assume that we know the pdf of $\mu$ which is doubly differentiable and referred to as $f(x)$. We first find $m$ such that $f'(m) = 0$. After, we use the second-order Taylor expansion of $\log f(x)$ around $m$:

$$\log f(x) \approx \log f(m) - \frac{1}{2\sigma^2}(x - m)^2,$$

where the first order does not appear since $f'(m) = 0$ and $\sigma^2 = \frac{-1}{\frac{d^2}{dx^2}\log f(x)\big|_{x=m}}$.

**Sampling-based approximation.** We are interested in approximating a continuous distribution $\mu$ by an Gaussian $\mathcal{N}(m, \sigma^2)$. Here, we assume that we do not know the pdf of $\mu$, however, we can sample from $X_1, \ldots, X_n \overset{i.i.d}{\sim} \mu$. Therefore, we can approximate $\mu$ by $\mathcal{N}(m, \sigma^2)$ with $m = \frac{1}{n}\sum_{i=1}^n X_i$ and $\sigma^2 = \frac{1}{n}\sum_{i=1}^n (X_i - m)^2$ which is equivalent to doing maximum likelihood estimate, moment matching, and doing optimization in Equation 10 with the proxy empirical measure $\mu_n = \frac{1}{n}\sum_{i=1}^n \delta_{X_i}$.

## B  NON-PARAMETRIC HYPOTHESIS TESTING

We follow the non-parametric two-sample test in (Wang et al., 2022). We use the permutation test for two cases. The first case is with two measures as Gaussians with diagonal covariance matrix and the second case is with two measures as Gaussians with full covariance matrix. We use the same hyper-parameters setting as in Section 5.1 in (Wang et al., 2022) and use SW, LCV-SW, and UCV-SW with $L = 100$ as the testing distance. We obtain the result in Figure 4.

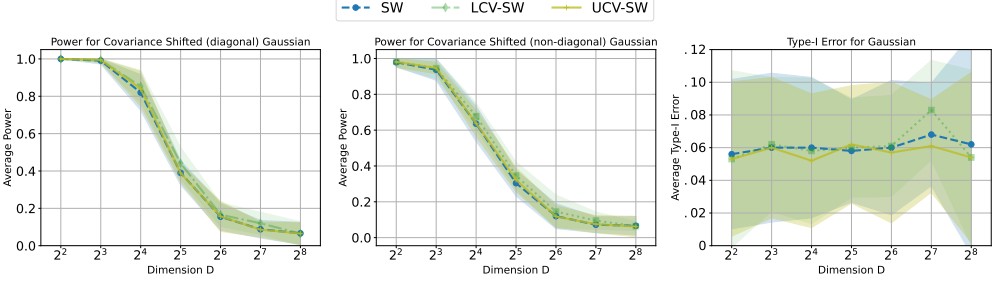

Figure 4: Testing results on Gaussian distributions across different choices of dimension D. Left: power for Gaussian distributions, where the shifted covariance matrix is still diagonal; Middle: power for Gaussian distributions, where the shifted covariance matrix is non-diagonal; Right: Type-I error.

## C  PROOFS

### C.1  PROOF OF PROPOSITION 1

We have $\mu$ and $\nu$ are two empirical measures i.e., $\mu = \sum_{i=1}^n \alpha_i \delta_{x_i}$ $(\sum_{i=1}^n \alpha_i = 1)$ and $\nu = \sum_{i=1}^m \beta_i \delta_{y_i}$ $(\sum_{i=1}^m \beta_i = 1)$, we have their projected measures $\theta\sharp\mu = \sum_{i=1}^n \alpha_i \delta_{\theta^\top x_i}$ and $\theta\sharp\nu = $

$\sum_{i=1}^{n} \beta_i \delta_{\theta^{\top} y_i}$. Now, we have:

$$\text{argmax}_{m_1, \sigma_1^2} \left[ \sum_{i=1}^{n} \alpha_i \log \left( \frac{1}{\sqrt{2\pi\sigma_1^2}} \exp \left( -\frac{1}{2\sigma_1^2} (\theta^{\top} x_i - m_1)^2 \right) \right) \right]$$

$$= \text{argmax}_{m_1, \sigma_1^2} \left[ \sum_{i=1}^{n} \left( -\frac{\alpha_i}{2\sigma_1^2} (\theta^{\top} x_i - m_1)^2 - \frac{\alpha_i}{2} \log(\sigma_1^2) \right) \right]$$

$$= \text{argmax}_{m_1, \sigma_1^2} f(m_1, \sigma_1^2).$$

Taking the derivatives and setting them to 0, we have:

$$\frac{d}{dm_1} f(m_1, \sigma_1^2) = \sum_{i=1}^{n} \frac{\alpha_i}{\sigma_1^2} (m_1 - \theta^{\top} x_i) = \frac{1}{\sigma_1^2} \left( \sum_{i=1}^{n} \alpha_i m_1 - \sum_{i=1}^{n} \alpha_i \theta^{\top} x_i \right) = 0,$$

$$\frac{d}{d\sigma_1^2} f(m_1, \sigma_1^2) = \sum_{i=1}^{n} \left( \frac{\alpha_i}{2\sigma_1^4} (\theta^{\top} x_i - m_1)^2 - \frac{\alpha_i}{2\sigma_1^2} \right) = \frac{1}{2\sigma_1^2} \left( \sum_{i=1}^{n} \frac{\alpha_i}{\sigma_1^2} (\theta^{\top} x_i - m_1)^2 - \sum_{i=1}^{n} \alpha_i \right) = 0.$$

Hence, we obtain:

$$m_1 = \frac{\sum_{i=1}^{n} \alpha_i \theta^{\top} x_i}{\sum_{i=1}^{n} \alpha_i} = \sum_{i=1}^{n} \alpha_i \theta^{\top} x_i,$$

$$\sigma_1^2 = \frac{\sum_{i=1}^{n} \alpha_i (\theta^{\top} x_i - m_1)^2}{\sum_{i=1}^{n} \alpha_i} = \sum_{i=1}^{n} \alpha_i (\theta^{\top} x_i - m_1)^2.$$

With similar derivation, we obtain the result for $m_2, \sigma_2^2$ and complete the proof.

### C.2 PROOF OF PROPOSITION 2

We first prove the following lemma.

**Lemma 1.** *Let $\theta$ is a uniformely distributed vector on the unit-hypersphere i.e., $\theta \sim \mathcal{U}(\mathbb{S}^{d-1})$, we have $\mathbb{E}[\theta\theta^{\top}] = \frac{1}{d} \mathbf{I}_d$.*

*Proof.* Since we can obtain a uniformly distributed vector $\theta$ on $\mathbb{S}^{d-1}$ by normalizing a unit Gaussian distributed vector i.e., $\theta = \frac{z}{||z||_2}$ with $z \sim \mathcal{N}(0, \mathbf{I}_d)$. Therefore, we can rewrite:

$$\mathbb{E}[\theta\theta^{\top}] = \mathbb{E} \left[ \frac{z}{||z||_2} \frac{z}{||z||_2}^{\top} \right].$$

Since the expectation of a random matrix is a matrix of expectation, we now need to calculate $\mathbb{E}[\theta_i \theta_j]$ or $\mathbb{E} \left[ \frac{z_i}{||z||_2} \frac{z_j}{||z||_2}^{\top} \right]$ for all $i, j \in [[d]]$.

Let denote $\theta = (\theta_1, \ldots, \theta_d)$ and $z = (z_1, \ldots, z_d)$, we have $\theta' = (\theta_1, \ldots, -\theta_i, \ldots, \theta_d)$ (for any $i \in [[d]] := \{1, 2, \ldots, d\}$ ) also follow $\mathcal{U}(\mathbb{S}^{d-1})$ since $\theta' = \frac{z'}{||z'||_2}$ with $z' = (z_1, \ldots, -z_i, \ldots, z_d) \sim \mathcal{N}(0, \mathbf{I}_d)$ (linear mapping of a Gaussian is a Gaussian). Therefore, we have $\mathbb{E}[\theta_i \theta_j] = -\mathbb{E}[\theta_i \theta_j]$ which means $\mathbb{E}[\theta_i \theta_j] = 0$ for all $j \in [[d]] \neq i$. Now, we calculate the variance $\mathbb{E}[\theta_i^2]$ for all $i \in [[d]]$. We have

$$\sum_{i=1}^{d} \mathbb{E}[\theta_i^2] = \sum_{i=1}^{d} \mathbb{E} \left[ \frac{z_i^2}{||z||_2^2} \right] = \sum_{i=1}^{d} \mathbb{E} \left[ \frac{z_i^2}{\sum_{j=1}^{d} z_j^2} \right] = 1.$$

Moreover, $\theta_1, \ldots, \theta_d$ are exchangeable with a similar argument with the symmetry and linear mapping property of Gaussian distribution. Therefore, they are indentically distributed i.e., $\mathbb{E}[\theta_i^2] = \mathbb{E}[\theta_j^2] = 1/d$ for all $i, j \in [[d]]$. We complete the proof here. $\square$

We now go back to the proof of Proposition 2. From Proposition 1, we have:

$$\mathbb{E}[(m_1(\theta;\mu) - m_2(\theta;\nu))^2] = \mathbb{E}\left[\left(\sum_{i=1}^{n}\alpha_i\theta^\top x_i - \sum_{i=1}^{m}\beta_i\theta^\top y_i\right)^2\right]$$

$$= \left(\sum_{i=1}^{n}\alpha_i x_i - \sum_{i=1}^{m}\beta_i y_i\right)^\top \mathbb{E}\left[\theta\theta^\top\right]\left(\sum_{i=1}^{n}\alpha_i x_i - \sum_{i=1}^{m}\beta_i y_i\right) = \frac{1}{d}\left\|\sum_{i=1}^{n}\alpha_i x_i - \sum_{i=1}^{m}\beta_i y_i\right\|^2,$$

where we use the result of $\mathbb{E}_{\theta\sim\mathcal{U}(\mathbb{S}^{d-1})}[\theta\theta^\top] = \frac{1}{d}$ from Lemma 1. Next, we have:

$$\mathbb{E}\left[\sigma_1^2(\theta;\mu)\right] = \mathbb{E}\left[\sum_{i=1}^{n}\alpha_i\left(\theta^\top x_i - \sum_{i'=1}^{n}\alpha_{i'}\theta^\top x_{i'}\right)^2\right]$$

$$= \sum_{i=1}^{n}\alpha_i\left(x_i - \sum_{i'=1}^{n}\alpha_{i'}x_{i'}\right)^\top \mathbb{E}\left[\theta\theta^\top\right]\left(x_i - \sum_{i'=1}^{n}\alpha_{i'}x_{i'}\right) = \frac{1}{d}\sum_{i=1}^{n}\alpha_i\left\|x_i - \sum_{i'=1}^{n}\alpha_{i'}x_{i'}\right\|^2.$$

Similarly, we have $\mathbb{E}\left[\sigma_2^2(\theta;\nu)\right] = \frac{1}{d}\sum_{i=1}^{m}\beta_i\left\|y_i - \sum_{i'=1}^{m}\beta_{i'}y_{i'}\right\|^2$.

## C.3 PROOF OF PROPOSITION 3

We recall that:

$C_{low}(\theta;\mu,\nu) = (m_1(\theta;\mu) - m_2(\theta;\nu))^2,$

$C_{up}(\theta;\mu,\nu) = (m_1(\theta;\mu) - m_2(\theta;\nu))^2 + \sigma_1(\theta;\mu)^2 + \sigma_2(\theta;\nu)^2,$

$W_2^2(\mathcal{N}(m_1(\theta;\mu),\sigma_1^2(\theta;\mu)),\mathcal{N}(m_2(\theta;\nu),\sigma_2^2(\theta;\nu))) = (m_1(\theta;\mu) - m_2(\theta;\nu))^2 + (\sigma_1(\theta;\mu) - \sigma_2(\theta;\nu))^2.$

Since $0 \le (\sigma_1(\theta;\mu) - \sigma_2(\theta;\nu))^2 \le \sigma_1(\theta;\mu)^2 + \sigma_2(\theta;\nu)^2$, we obtain the result by adding $(m_1(\theta;\mu) - m_2(\theta;\nu))^2$ to the inequalities.

## C.4 PROOF OF PROPOSITION 4

We have $\mu$ and $\nu$ are two empirical measures i.e., $\mu = \sum_{i=1}^{n}\alpha_i\delta_{x_i}$ ($\sum_{i=1}^{n}\alpha_i = 1$) and $\nu = \sum_{i=1}^{m}\beta_i\delta_{y_i}$ ($\sum_{i=1}^{m}\beta_i = 1$). Now, we have:

$$\text{argmax}_{\mathbf{m_1},\sigma_1^2}\sum_{i=1}^{n}\alpha_i\log\left(\frac{1}{\sqrt{(2\pi)^d|\sigma_1^2\mathbf{I}|}}\exp\left(-\frac{1}{2}(x_i - \mathbf{m_1})^\top|\sigma_1^2\mathbf{I}|^{-1}(x_i - \mathbf{m_1})\right)\right)$$

$$= \text{argmax}_{\mathbf{m_1},\sigma_1^2}\sum_{i=1}^{n}\alpha_i\left(-\frac{1}{2\sigma_1^{2d}}(x_i - \mathbf{m_1})^\top(x_i - \mathbf{m_1}) - \frac{d}{2}\log(\sigma_1^2)\right)$$

$$= \text{argmax}_{\mathbf{m_1},\sigma_1^2}f(\mathbf{m_1},\sigma_1^2)$$

Taking the derivatives and setting them to 0, we have:

$$\nabla_{\mathbf{m_1}}f(\mathbf{m_1},\sigma_1^2) = \sum_{i=1}^{n}\frac{\alpha_i}{\sigma_1^{2d}}(\mathbf{m_1} - x_i) = \frac{1}{\sigma_1^{2d}}\left(\sum_{i=1}^{n}\alpha_i\mathbf{m_1} - \alpha_i x_i\right) = 0,$$

$$\frac{d}{d\sigma_1^2}f(\mathbf{m_1},\sigma_1^2) = \sum_{i=1}^{n}\alpha_i\left(\frac{d}{2\sigma_1^{2d+2}}(x_i - \mathbf{m_1})^\top(x_i - \mathbf{m_1}) - \frac{d}{2\sigma_1^2}\right),$$

$$= \frac{d}{d\sigma_1^2}\sum_{i=1}^{n}\left(\frac{\alpha_i\|x_i - \mathbf{m_1}\|_2^2}{\sigma_1^{2d}} - \alpha_i\right) = 0.$$

Hence, we obtain:

$$\mathbf{m_1} = \frac{\sum_{i=1}^{n} \alpha_i x_i}{\sum_{i=1}^{n} \alpha_i} = \sum_{i=1}^{n} \alpha_i x_i,$$

$$\sigma_1^2 = \left( \frac{\sum_{i=1}^{n} \alpha_i \|x_i - \mathbf{m_1}\|_2^2}{\sum_{i=1}^{n} \alpha_i} \right)^{\frac{1}{d}} = \left( \sum_{i=1}^{n} \alpha_i \|x_i - \mathbf{m_1}\|_2^2 \right)^{\frac{1}{d}}.$$

Similarly, we obtain $\mathbf{m_2} = \sum_{i=1}^{m} \beta_i y_i$ and $\sigma_2^2 = \left( \sum_{i=1}^{m} \beta_i \|y_i - \mathbf{m_2}\|_2^2 \right)^{\frac{1}{d}}$.

Using the linearity of Gaussian distributions, we have $\theta \sharp \mathcal{N}(\mathbf{m_1}, \sigma_1^2 \mathbf{I}) = \mathcal{N}(\theta^\top \mathbf{m_1}, \sigma_1^2 \theta^\top \mathbf{I} \theta) = \mathcal{N}(\theta^\top \mathbf{m_1}, \sigma_1^2)$ $(\theta^\top \theta = 1)$. Similarly, we obtain $\theta \sharp \mathcal{N}(\mathbf{m_2}, \sigma_2^2 \mathbf{I}) = \mathcal{N}(\theta^\top \mathbf{m_2}, \sigma_2^2)$. Therefore, we have:

$$\mathrm{W}_2^2(\theta \sharp \mathcal{N}(\mathbf{m_1}, \sigma_1^2 \mathbf{I}), \theta \sharp \mathcal{N}(\mathbf{m_2}, \sigma_2^2 \mathbf{I})) = (\theta^\top \mathbf{m_1} - \theta^\top \mathbf{m_2})^2 + (\sigma_1 - \sigma_2)^2.$$

Calculating the expectation, we have:

$$\mathbb{E}\left[ \mathrm{W}_2^2(\theta \sharp \mathcal{N}(\mathbf{m_1}, \sigma_1^2 \mathbf{I}), \theta \sharp \mathcal{N}(\mathbf{m_2}, \sigma_2^2 \mathbf{I})) \right] = (\mathbf{m_1} - \mathbf{m_2})^\top \mathbb{E}[\theta \theta^\top](\mathbf{m_1} - \mathbf{m_2}) + (\sigma_1 - \sigma_2)^2$$

$$= \frac{1}{d} \|\mathbf{m_1} - \mathbf{m_2}\|_2^2 + (\sigma_1 - \sigma_2)^2.$$

Let denote $C(\theta) = \mathrm{W}_2^2(\theta \sharp \mathcal{N}(\mathbf{m_1}, \sigma_1^2 \mathbf{I}), \theta \sharp \mathcal{N}(\mathbf{m_2}, \sigma_2^2 \mathbf{I}))$, we have:

$$C(\theta) - \mathbb{E}[C(\theta)] = (\theta^\top \mathbf{m_1} - \theta^\top \mathbf{m_2})^2 - \frac{1}{d} \|\mathbf{m_1} - \mathbf{m_2}\|_2^2$$

$$= \left( \theta^\top \sum_{i=1}^{n} \alpha_i x_i - \theta^\top \sum_{i=1}^{m} \beta_i y_i \right)^2 - \frac{1}{d} \left\| \sum_{i=1}^{n} \alpha_i x_i - \sum_{i=1}^{m} \beta_i y_i \right\|_2^2$$

$$= \left( \sum_{i=1}^{n} \alpha_i \theta^\top x_i - \sum_{i=1}^{m} \beta_i \theta^\top y_i \right)^2 - \frac{1}{d} \left\| \sum_{i=1}^{n} \alpha_i x_i - \sum_{i=1}^{m} \beta_i y_i \right\|_2^2$$

$$= C_{low}(\theta; \mu, \nu) - \mathbb{E}[C_{low}(\theta; \mu, \nu)].$$

Similarly, we obtain:

$$\mathrm{Var}\left[ C(\theta) \right] = \mathbb{E}\left[ (C(\theta) - \mathbb{E}[C(\theta)])^2 \right] = \mathbb{E}\left[ (C_{low}(\theta; \mu, \nu) - \mathbb{E}[C_{low}(\theta; \mu, \nu)])^2 \right] = \mathrm{Var}[C_{low}(\theta; \mu, \nu)],$$

$$\mathrm{Cov}\left[ C(\theta), W(\theta; \mu, \nu) \right] = \mathbb{E}\left[ (C(\theta) - \mathbb{E}[C(\theta)])(W(\theta; \mu, \nu) - \mathbb{E}[W(\theta; \mu, \nu)]) \right]$$

$$= \mathbb{E}\left[ (C_{low}(\theta; \mu, \nu) - \mathbb{E}[C_{low}(\theta; \mu, \nu)])(W(\theta; \mu, \nu) - \mathbb{E}[W(\theta; \mu, \nu)]) \right]$$

$$= \mathrm{Cov}\left[ C_{low}(\theta; \mu, \nu), W(\theta; \mu, \nu) \right].$$

Therefore, it is sufficient to claim that using the $C_{low}$ control variate is equivalent to use $\mathrm{W}_2^2(\theta \sharp \mathcal{N}(\mathbf{m_1}, \sigma_1^2 \mathbf{I}), \theta \sharp \mathcal{N}(\mathbf{m_2}, \sigma_2^2 \mathbf{I}))$ as the control variate.

# D  ALGORITHMS

We present the algorithm for computing the conventional Monte Carlo estimator of the sliced Wasserstein distance between two discrete measures in Algorithm 1. Similarly, we provide the algorithms for the lower bound control variate estimator and the upper bound control variate estimator in Algorithm 2 and in Algorithm 3 respectively.

# E  RELATED WORKS

**Sliced Wasserstein variants with different projecting functions.** The conventional sliced Wasserstein is based on a linear projecting function i.e., the inner product to project measures to one dimension. In some special cases of probability measures, some other projecting functions might be preferred e.g., generalized sliced Wasserstein (Kolouri et al., 2019) distance with circular, polynomial projecting function, spherical sliced Wasserstein (Bonet et al., 2023) with geodesic spherical projecting function, and so on. Despite having different projecting functions, all mentioned sliced

---

**Algorithm 1** The conventional estimator of sliced Wasserstein distance.

---

**Input:** Probability measures $\mu = \sum_{i=1}^{n} \alpha_i \delta_{x_i}$ and $\nu = \sum_{i=1}^{m} \beta_i \delta_{y_i}$, $p \geq 1$, and the number of projections $L$.
Set $\widehat{\mathrm{SW}}_p^p(\mu, \nu; L) = 0$
**for** $l = 1$ to $L$ **do**
    Sample $\theta_l \sim \mathcal{U}(\mathbb{S}^{d-1})$
    Compute $\widehat{\mathrm{SW}}_p^p(\mu, \nu; L) = \widehat{\mathrm{SW}}_p^p(\mu, \nu; L) + \frac{1}{L} \int_0^1 |F_{\theta_l \sharp \mu}^{-1}(z) - F_{\theta_l \sharp \nu}^{-1}(z)|^p dz$
**end for**
**Return:** $\widehat{\mathrm{SW}}_p^p(\mu, \nu; L)$

---

**Algorithm 2** The lower bound control variate estimator of sliced Wasserstein distance.

---

**Input:** Probability measures $\mu = \sum_{i=1}^{n} \alpha_i \delta_{x_i}$ and $\nu = \sum_{i=1}^{m} \beta_i \delta_{y_i}$, $p \geq 1$, and the number of projections $L$.
Set $\widehat{\mathrm{SW}}_p^p(\mu, \nu; L) = 0$
Compute $\bar{x} = \sum_{i=1}^{n} \alpha_i x_i$, and $\bar{y} = \sum_{i=1}^{m} \beta_i y_i$
**for** $l = 1$ to $L$ **do**
    Sample $\theta_l \sim \mathcal{U}(\mathbb{S}^{d-1})$
    Compute $w_l = \int_0^1 |F_{\theta_l \sharp \mu}^{-1}(z) - F_{\theta_l \sharp \nu}^{-1}(z)|^p dz$
    Compute $c_l = (\theta^\top \bar{x} - \theta^\top \bar{y})^2$
**end for**
Compute $\widehat{\mathrm{SW}}_p^p(\mu, \nu; L) = \frac{1}{L} \sum_{l=1}^{L} w_l$
Compute $b = \frac{1}{d} \|\bar{x} - \bar{y}\|_2^2$
Compute $\gamma = \frac{\frac{1}{L} \sum_{l=1}^{L} (w_l - \widehat{\mathrm{SW}}_p^p(\mu, \nu; L))(c_l - b)}{\frac{1}{L} \sum_{l=1}^{L} (c_l - b)^2}$
Compute $\widehat{\mathrm{LCV\text{-}SW}}_p^p(\mu, \nu; L) = \widehat{\mathrm{SW}}_p^p(\mu, \nu; L) - \gamma \frac{1}{L} \sum_{l=1}^{L} (c_l - b)$
**Return:** $\widehat{\mathrm{LCV\text{-}SW}}_p^p(\mu, \nu; L)$

---

**Algorithm 3** The upper bound control variate estimator of sliced Wasserstein distance.

---

**Input:** Probability measures $\mu = \sum_{i=1}^{n} \alpha_i \delta_{x_i}$ and $\nu = \sum_{i=1}^{m} \beta_i \delta_{y_i}$, $p \geq 1$, and the number of projections $L$.
Set $\widehat{\mathrm{SW}}_p^p(\mu, \nu; L) = 0$
Compute $\bar{x} = \sum_{i=1}^{n} \alpha_i x_i$, and $\bar{y} = \sum_{i=1}^{m} \beta_i y_i$
**for** $l = 1$ to $L$ **do**
    Sample $\theta_l \sim \mathcal{U}(\mathbb{S}^{d-1})$
    Compute $w_l = \int_0^1 |F_{\theta_l \sharp \mu}^{-1}(z) - F_{\theta_l \sharp \nu}^{-1}(z)|^p dz$
    Compute $c_l = (\theta^\top \bar{x} - \theta^\top \bar{y})^2 + \sum_{i=1}^{n} \alpha_i (\theta^\top x_i - \theta^\top \bar{x})^2 + \sum_{i=1}^{m} \beta_i (\theta^\top y_i - \theta^\top \bar{y})^2$
**end for**
Compute $\widehat{\mathrm{SW}}_p^p(\mu, \nu; L) = \frac{1}{L} \sum_{l=1}^{L} w_l$
Compute $b = \frac{1}{d} \|\bar{x} - \bar{y}\|_2^2 + \frac{1}{d} \sum_{i=1}^{n} \alpha_i \|(x_i - \bar{x})\|_2^2 + \frac{1}{d} \sum_{i=1}^{m} \beta_i \|(y_i - \bar{y})\|_2^2$
Compute $\gamma = \frac{\frac{1}{L} \sum_{l=1}^{L} (w_l - \widehat{\mathrm{SW}}_p^p(\mu, \nu; L))(c_l - b)}{\frac{1}{L} \sum_{l=1}^{L} (c_l - b)^2}$
Compute $\widehat{\mathrm{UCV\text{-}SW}}_p^p(\mu, \nu; L) = \widehat{\mathrm{SW}}_p^p(\mu, \nu; L) - \gamma \frac{1}{L} \sum_{l=1}^{L} (c_l - b)$
**Return:** $\widehat{\mathrm{UCV\text{-}SW}}_p^p(\mu, \nu; L)$

---

Wasserstein variants utilize random projecting directions that follow the uniform distribution over the unit hypersphere and are estimated using the Monte Carlo integration scheme. Therefore, we could directly adapt the proposed control variates in these variants.

**Applications of the control variate estimators.** Since the proposed control variates are used to estimate the SW distance, they can be applied to all applications where sliced Wasserstein exists. We would like to mention some other applications such as domain adaptation (Lee et al., 2019),

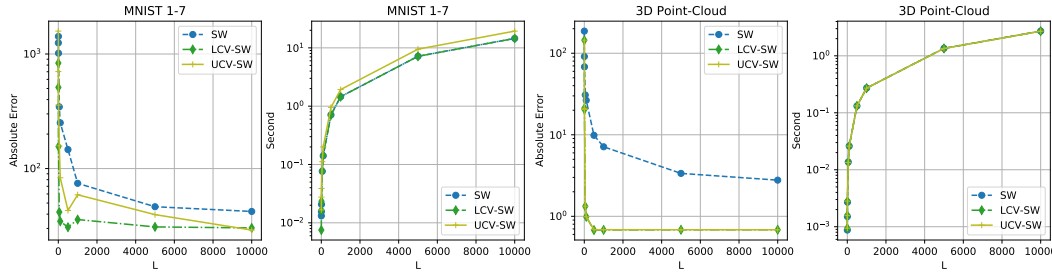

Figure 5: The empirical errors of the conventional estimator (SW) and the control variate estimators (LCV-SW, UCV-SW) when comparing empirical distributions over MNIST images and point-clouds.

Table 3: Estimated variances of the conventional estimator's and the control variate estimators.

| Estimator | MNIST 0-1 | MNIST 1-7 | Point-cloud-1 | Point-cloud-2 |
|---|---|---|---|---|
| SW | $4700.87 \pm 59.12$ | $1205.62 \pm 14.17$ | $12.78 \pm 0.025$ | $12.79 \pm 0.026$ |
| LCV-SW | $\mathbf{0.045 \pm 0.008}$ | $\mathbf{0.0017 \pm 0.001}$ | $\mathbf{0.0025 \pm 0.0000}$ | $\mathbf{0.0021 \pm 0.0000}$ |
| UCV-SW | $0.061 \pm 0.016$ | $0.0018 \pm 0.0001$ | $\mathbf{0.0025 \pm 0.0000}$ | $\mathbf{0.0021 \pm 0.0000}$ |

Table 4: Summary of Wasserstein-2 scores (multiplied by $10^4$) from 3 different runs, computational time in second (s) to reach step 500 of different sliced Wasserstein variants in gradient flows.

| Distances | Step 3000 ($W_2 \downarrow$) | Step 4000 ($W_2 \downarrow$) | Step 5000 ($W_2 \downarrow$) | Step 6000($W_2 \downarrow$) | Step 8000 ($W_2 \downarrow$) | Time (s $\downarrow$) |
|---|---|---|---|---|---|---|
| SW L=10 | $305.3907 \pm 0.4919$ | $137.7762 \pm 0.3630$ | $36.1807 \pm 0.1383$ | $0.1054 \pm 0.0022$ | $2.3e-5 \pm 1.0e-5$ | $\mathbf{26.30 \pm 0.03}$ |
| LCV-SW L=10 | $302.9718 \pm 0.1788$ | $\mathbf{135.9132 \pm 0.0922}$ | $35.0292 \pm 0.1457$ | $0.0452 \pm 0.0045$ | $\mathbf{1.7e-5 \pm 0.3e-5}$ | $27.65 \pm 0.01$ |
| UCV-SW L=10 | $\mathbf{302.9717 \pm 0.1788}$ | $\mathbf{135.9132 \pm 0.0922}$ | $35.0295 \pm 0.1458$ | $\mathbf{0.0446 \pm 0.0038}$ | $2.0e-5 \pm 0.4e-5$ | $29.56 \pm 0.01$ |
| SW L=100 | $300.6303 \pm 0.2375$ | $134.0492 \pm 0.3146$ | $33.8608 \pm 0.1348$ | $0.0121 \pm 0.0010$ | $1.6e-5 \pm 0.2e-5$ | $\mathbf{222.06 \pm 1.34}$ |
| LCV-SW L=100 | $300.2362 \pm 0.0054$ | $133.5238 \pm 0.0065$ | $33.4460 \pm 0.0030$ | $0.0084 \pm 5.8e-5$ | $\mathbf{1.4e-5 \pm 0.1e-5}$ | $223.79 \pm 0.82$ |
| UCV-SW L=100 | $\mathbf{300.2631 \pm 0.0054}$ | $\mathbf{133.5237 \pm 0.0065}$ | $\mathbf{33.4459 \pm 0.0030}$ | $\mathbf{0.0083 \pm 8.5e-5}$ | $1.6e-5 \pm 0.1e-5$ | $235.29 \pm 1.65$ |

approximate Bayesian computation (Nadjahi et al., 2020a), color transfer (Li et al., 2022), point-cloud reconstruction (Nguyen et al., 2024a), mesh deformation Le et al. (2024a), diffusion models (Nguyen et al., 2024b), and many other tasks.

# F ADDITIONAL EXPERIMENTS

In this section, we provide some additional experiments for applications in the main text. In particular, we calculate the empirical variances of the conventional estimator and the control variate estimators on different images of digits and point-clouds in Appendix F.1. We provide the point-cloud gradient flow between two new point-cloud in Appendix F.2. We then provide the detailed training of deep generative models and additional generated images in Appendix F.3.

## F.1 COMPARING EMPIRICAL PROBABILITY MEASURES OVER IMAGES AND POINT-CLOUDS

**Settings.** We follow the same settings which are used in the main text. However, for MNIST, we compare probability measures over digit 1 and digit 7. For point-clouds, we compare two different point-clouds from the main text. We report the estimated variances of $W(\theta; \mu, \nu), Z_{low}(\theta; \mu, \nu), Z_{up}(\theta; \mu, \nu)$ defined in Definition 4 in Table 3. We use a large number of samples e.g., 100000 Monte Carlo samples. In the table, point-cloud-1 denotes the pair in Figure 2 and point-cloud-2 denotes the pair in Figure 6.

**Results.** We show the estimated errors and the corresponding computational time in Figure 5. From the figure, we observe the same phenomenon as in the main text. In particular, the control variate estimators reduce considerably the errors in both cases while having approximately the same computational time.

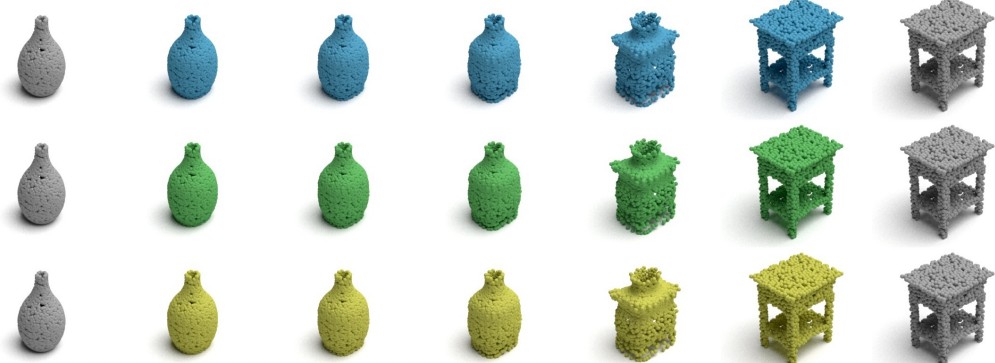

Figure 6: Point-Cloud gradient flows for $L = 10$ from SW, LCV-SW, and UCV-SW.

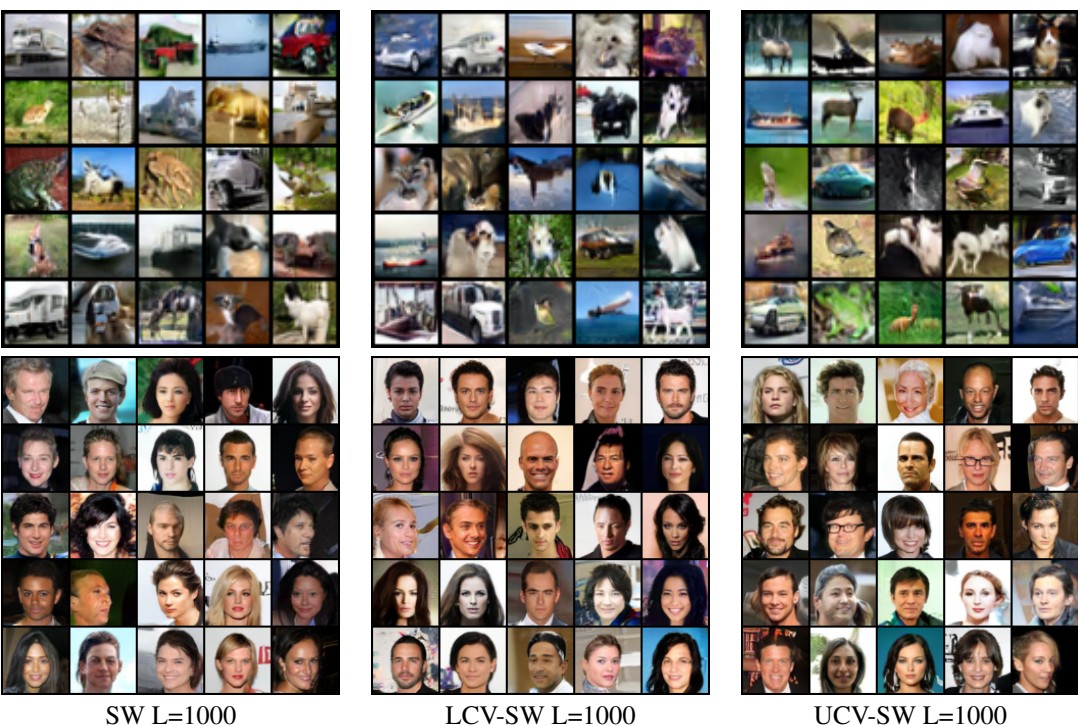

| SW L=1000 | LCV-SW L=1000 | UCV-SW L=1000 |

Figure 7: Random generated images of distances on CIFAR10 and CelebA.

## F.2 POINT-CLOUD GRADIENT FLOWS

**Settings.** We follow the same settings which are used in the main text. However, we use different point-clouds which are also used in Appendix F.1.

**Results.** We show the quantitative results in Table 4 and the corresponding qualitative result in Figure 6. Overall, we observe the same phenomenon as in the main text. In particular, the control variate estimators i.e., LCV-SW, UCV-SW help to drive the flows to converge faster to the target point-cloud than the conventional estimator i.e., SW. It is worth noting that the computational time of the control variate estimators is only slightly higher than the conventional estimator for both settings of the number of projections $L = 10, 100$.

## F.3 DEEP GENERATIVE MODELING

**Setting.** We denote $\mu$ as our data distribution. We then design the model distribution $\nu_\phi$ as a push forward probability measure that is created by pushing a unit multivariate Gaussian ($\epsilon$) through a

neural network $G_\phi$ that maps from the realization of the noise to the data space. We use a second neural network $T_\beta$ that maps from data space to a single scalar. We denote $T_{\beta_1}$ is the sub neural network of $T_\beta$ that maps from the data space to a feature space (output of the last Resnet block), and $T_{\beta_2}$ that maps from the feature space (image of $T_{\beta_1}$) to a single scalar. More precisely, $T_\beta = T_{\beta_2} \circ T_{\beta_1}$. We use the following neural networks for $G_\phi$ and $T_\beta$:

- **CIFAR10**:
  - $G_\phi$: $z \in \mathbb{R}^{128}(\sim \epsilon : \mathcal{N}(0,1)) \to 4 \times 4 \times 256$(Dense, Linear) $\to$ ResBlock up 256 $\to$ ResBlock up 256 $\to$ ResBlock up 256 $\to$ BN, ReLU, $\to 3 \times 3$ conv, 3 Tanh .
  - $T_{\beta_1}$: $x \in [-1,1]^{32 \times 32 \times 3} \to$ ResBlock down 128 $\to$ ResBlock down 128 $\to$ ResBlock down 128 $\to$ ResBlock 128 $\to$ ResBlock 128.
  - $T_{\beta_2}$: $\boldsymbol{x} \in \mathbb{R}^{128 \times 8 \times 8} \to$ ReLU $\to$ Global sum pooling(128) $\to$ 1(Spectral normalization).
  - $T_\beta(x) = T_{\beta_2}(T_{\beta_1}(x))$.
- **CelebA:**
  - $G_\phi$: $z \in \mathbb{R}^{128}(\sim \epsilon : \mathcal{N}(0,1)) \to 4 \times 4 \times 256$(Dense, Linear) $\to$ ResBlock up 256 $\to$ ResBlock up 256 $\to$ ResBlock up 256 $\to$ ResBlock up 256 $\to$ BN, ReLU, $\to 3 \times 3$ conv, 3 Tanh .
  - $T_{\beta_1}$: $\boldsymbol{x} \in [-1,1]^{32 \times 32 \times 3} \to$ ResBlock down 128 $\to$ ResBlock down 128 $\to$ ResBlock down 128 $\to$ ResBlock 128 $\to$ ResBlock 128.
  - $T_{\beta_2}$: $\boldsymbol{x} \in \mathbb{R}^{128 \times 8 \times 8} \to$ ReLU $\to$ Global sum pooling(128) $\to$ 1(Spectral normalization).
  - $T_\beta(x) = T_{\beta_2}(T_{\beta_1}(x))$.

We use the following bi-optimization problem to train our neural networks:

$$\min_{\beta_1, \beta_2} \left( \mathbb{E}_{x \sim \mu}[\min(0, -1 + T_\beta(x))] + \mathbb{E}_{z \sim \epsilon}[\min(0, -1 - T_\beta(G_\phi(z)))] \right),$$

$$\min_\phi \mathbb{E}_{X \sim \mu^{\otimes m}, Z \sim \epsilon^{\otimes m}}[\mathcal{S}(\tilde{T}_{\beta_1,\beta_2} \sharp P_X, \tilde{T}_{\beta_1,\beta_2} \sharp G_\phi \sharp P_Z)],$$

where the function $\tilde{T}_{\beta_1,\beta_2} = [T_{\beta_1}(x), T_{\beta_2}(T_{\beta_1}(x))]$ which is the concatenation vector of $T_{\beta_1}(x)$ and $T_{\beta_2}(T_{\beta_1}(x))$, $\mathcal{S}$ is an estimator of the sliced Wasserstein distance. The number of training iterations is set to 100000 on CIFAR10 and 50000 in CelebA. We update the generator $G_\phi$ every 5 iterations and we update the feature function $T_\beta$ every iteration. The mini-batch size $m$ is set to 128 in all datasets. We use the Adam (Kingma & Ba, 2014) optimizer with parameters $(\beta_1, \beta_2) = (0, 0.9)$ for both $G_\phi$ and $T_\beta$ with the learning rate 0.0002. We use 50000 random samples from estimated generative models $G_\phi$ for computing the FID scores and the Inception scores. In evaluating FID scores, we use all training samples for computing statistics of datasets.

**Results.** In addition to the result in the main text, we provide generated images from the conventional estimator (SW) and the control variate estimator (LCV-SW and UCV-SW) with the number of projections $L = 1000$ in Figure 7. Overall, we see that by increasing the number of projections, the generated images are visually improved for all estimators. This result is consistent with the FID scores and the IS scores in Table 2.

## G  COMPUTATIONAL INFRASTRUCTURE

For comparing empirical probability measures over images and point-cloud application, and the point-cloud gradient flows application, we use a Macbook Pro M1 for conducting experiments. For deep generative modeling, experiments are run on a single NVIDIA V100 GPU.

