# OpenReview forum: "Sliced Wasserstein Estimation with Control Variates"
_ICLR.cc/2024/Conference — ICLR 2024 poster_

### Official Review · Reviewer_yEdx · 2023-10-27

**Soundness:** 3 good
**Presentation:** 3 good
**Contribution:** 2 fair
**Rating:** 6
**Confidence:** 4

**Summary:**

The paper focuses on improving the Monte Carlo estimation scheme used in computing Sliced Wasserstein (SW). While several works in the literature have concentrated on enhancing the sampling scheme, such as max-SW [1], projected W [2], and distributional SW [3], the paper's originality lies in its approach to controlling the variance of the estimation using control variates. The solution can be computed in linear time, similar to SW, and experiments demonstrate that, with a fixed number of lines L, the estimation is improved when compared to SW.

[1] Deshpande, Ishan, et al. "Max-sliced wasserstein distance and its use for gans." Proceedings of the IEEE/CVF Conference on Computer Vision and Pattern Recognition. 2019.

[2] Rowland, Mark, et al. "Orthogonal estimation of wasserstein distances." The 22nd International Conference on Artificial Intelligence and Statistics. PMLR, 2019.

[3] Nguyen, Khai, et al. "Distributional Sliced-Wasserstein and Applications to Generative Modeling." International Conference on Learning Representations. 2020.

**Strengths:**

The paper addresses an important problem: how to provide a proxy for the Wasserstein distance using a fast algorithm. It aims to take a step forward in SW estimation (which has a complexity of $O(n \log(n))$) by carefully selecting the projection lines. For a given number of directions $L$, the algorithms provide improved performance over SW, as evaluated in several experimental setups.
The originality of the approach is that it relies on the control variates method, which reduces variance in Monte Carlo methods. The paper is clear, and its claims are supported by empirical and/or theoretical evidence.

Main strengths:
- Fairly well motivated and original solution
- Theory and experiments seem sounded

**Weaknesses:**

The paper lacks positioning with respect to competitors whose goal is also to improve the sampling scheme of SW, or more generally, other SW variants. Evaluating the proposed methods among numerous competitors can be challenging. However, CV-SW is benchmarked only against SW, whereas algorithms such as max-SW (which considers only the 'max' direction), distributional SW (which searches for an 'optimal' direction), or even projected Wasserstein (which uses orthogonal directions) are closely related to the proposed method in spirit. Theoretical and experimental comparisons are missing.



Minor comments:
- $W(\theta)$ (two lines after eq. 7) has not been defined
- Could you check the value of $f'(\gamma)$ tow lines after eq. 8 ?

**Questions:**

- how does the method compares with max-sliced SW, distributional SW and projected SW?
- Is there any results that compare the value provided by CV-SW with SW or Wasserstein ? Do you have any results regarding the sample complexity?

---

> ### Author Response · Authors · 2023-11-16
> **Response to Reviewer yEdx**
>
> We would like to thank the reviewer for the insightful comments. We would like to discuss them as follow:
>
> **Q14**: The paper lacks positioning with respect to competitors whose goal is also to improve the sampling scheme of SW, or more generally, other SW variants. Evaluating the proposed methods among numerous competitors can be challenging. However, CV-SW is benchmarked only against SW, whereas algorithms such as max-SW (which considers only the 'max' direction), distributional SW (which searches for an 'optimal' direction), or even projected Wasserstein (which uses orthogonal directions) are closely related to the proposed method in spirit. Theoretical and experimental comparisons are missing.
>
> **A14**: Thank you very much for your insightful comments. Max sliced Wasserstein (Max-SW), distributional sliced Wasserstein (DSW), and Projected Wasserstein (PW) are variant of Sliced Wasserstein which focuses on changing the slicing distribution of SW from uniform distribution over the sphere to a different distribution (e.g., optimal Dirac delta measure, optimal von-Mises Fisher distribution, and so on). They are different distances that are different from SW i.e. different topological properties, different computational schemes. In contrast, we focus on the numerical approximation/estimation aspect of SW by reducing the variance of the estimator.  Our control variates are constructed specifically for the uniform slicing distribution.
>
> Although our control variate are constructed for the uniform distribution, they are still applicable for DSW, and PW by using importance sampling. In more detail, we can rewrite
> $
> DSW(\theta) = =  \mathbb{E}_{U}[W(\theta)\frac{\sigma(\theta)}{\sigma_0 (\theta)}],
> $
> For $\sigma(\theta)$ is the slicing distribution from DSW and $\sigma_0(\theta) = \mathcal{U}(\mathbb{S}^{d-1})$ is the uniform distribution. Therefore, we can use the proposed control variates to reduce the variance of the importance sampling. For PW, we can write it as:
>
> $
> PW(\theta) = =  \mathbb{E}_{\sigma_0(\theta_1)\ldots \sigma_0(\theta_K)}[W(\theta_1,\ldots,\theta_K)\frac{\sigma(\theta_1,\ldots,\theta_K))}{\sigma_0(\theta_1)\ldots \sigma_0(\theta_K)}],
> $
> For $\sigma(\theta_1,\ldots,\theta_K)$ is the orthogonal-based slicing distribution from PW. Therefore, we can still apply the control variates for PW.
>
>
> Finally, we would like to recall distances such as DSW, PW, and Max-SW are more computationally expensive than  SW. Therefore, SW has been still widely used in applications.
>
>
> **Q15**: how does the method compares with max-sliced SW, distributional SW and projected SW?
>
> **A15**: As discussed in Q14, SW is easy and stable to compute since it does not involve optimization as DSW, Max-SW, and orthogonal constraints as PW. From the statistical perspective, the uniform slicing distribution is suitable to use when we do not have any prior information about two compared distributions.  Moreover, LCV-SW and UCV-SW are estimations of SW which means that they are still SW.  The main aim of the paper is to bridge the literature on variance reduction and SW by proposing control variates for SW. These control variates can still be applied to DSW and PW as discussed.
>
> **Q16**:Is there any results that compare the value provided by CV-SW with SW or Wasserstein ?
>
>
> **A16**: We would like to recall that we do not propose a new SW metric. We propose new estimators for SW. Therefore, the interested distance is still SW.  CV-SW will asymptotically converge to  SW when increasing the number of Monte Carlo samples or the number of projections. Since the projecting direction belongs to the unit-hypersphere, we can show SW is a lower bound of Wasserstien distance. Therefore, CV-SW might be a lower-bound of Wasserstein distance with high probability.
>
> **Q17**: Do you have any results regarding the sample complexity?
>
> **A17**: We are still interested in SW which does not suffer from the curse of dimensionality and has a well-investigated sample complexity [R1] [R2] i.e., the expected value of SW between a measure and its corresponding empirical measure. Since CV-SW is an estimation of SW, the sample complexity has not been defined for CV-SW. For estimation, the interested quantity is an approximation error. As mentioned by **Reviewer pTc9**, the theoretical understanding of approximation error of using control variate is limited. However, it is widely recognized that using control variate is a practical method for improving the approximation error by reducing the variance.
>
> [R1] Statistical and Topological Properties of Sliced Probability Divergences
>
> [R2] Statistical, Robustness, and Computational Guarantees for Sliced Wasserstein Distances

---

> > ### Comment · Reviewer_yEdx · 2023-11-20
> >
> > Thank you for the comprehensive reply. I concur that the construction of SW with control variates differs from current SW variants.
> >
> > From my understanding, the main goal of CV-SW is to furnish a computationally fast and efficient estimator of SW for diverse applications, a task the paper successfully accomplishes. However, I still believe it falls short in clearly delineating its position in relation to other alternatives when comparing distributions using Wasserstein-like distances (even if I acknowledge that the experimental section of a paper doesn't need to encompass every possible competitor).
> >
> > I am unsure of what you mean by "PW, and Max-SW are more computationally expensive than SW". PW comes down to averaging over all couplings induced by the ordering on the projections on the line and hence enjoys similar computational complexity. Max-SW entails identifying the most meaningful projection direction among the possible set of lines.
> > Could you elaborate on your claim?

---

> ### Author Response · Authors · 2023-11-20
> **Response to Reviewer yEdx**
>
> We would like to thank the reviewer again for giving us additional feedback.
>
> We would like to clarify our claim of "PW, and Max-SW are more computationally expensive than SW".
>
> **For Max-SW**, a gradient ascent algorithm needs to be used to find the max projecting direction. To the best of our knowledge, there is no guarantee that the algorithm can converge to the global maxima. In practice, the algorithm is used with $T$ iterations. For each iteration, the gradient is computed based on the value of the one-dimensional Wasserstein distance, therefore the overall computational complexity is $\mathcal{O}(Tn\log n)$ with additional computation for gradient and update. When we set $T=L$ with $L$ as the number of projections in SW, Max-SW is slower than SW. We ran Max-SW with $T=10$ in the gradient flow application, and we got this result.
>
> |Estimators|  Step 3000  ($\text{W}_2\downarrow$)| Step 4000 ($\text{W}_2\downarrow$)| Step 5000  ($\text{W}_2\downarrow$)| Step 6000($\text{W}_2\downarrow$) | Step 8000 ($\text{W}_2\downarrow$)|Times (s)|
> |---------------|----------|----------|-----------|---------|------------|-------|
> Max-SW T=10| $39.66523\pm0.02$ | $0.06483\pm0.01$|$0.0062\pm0.0$|$0.00622\pm0.0$|$0.00583\pm0.0$|$60.52\pm1.21$
>
> We can see that Max-SW can make the gradient flow reduce the $W_2$ distance very fast in terms of the number of iterations. However, when compared in terms of real-time, it is much slower than SW (in Table 1). In addition, the flow from Max-SW cannot converge well (the $W_2$ score at the last iteration is higher than from SW) since it only uses one direction and can be seen as a minimax problem.
>
> **For PW**, the orthogonal constraint is enforced via the Gram-Smith orthogonality process (or QR decomposition) which has the computational complexity is $\mathcal{O}(L^2 d)$  for $L$ is the number of projections (quadratic). Moreover, the sorting operator is more optimized in Python than Gram-Smith orthogonality process in Pytorch to the best of our knowledge. Therefore, PW is slower than SW. Hence, for the same computational time, PW has a smaller number of projections than SW which might lead to worse performance. We ran $L=3$ for PW, and got this result:
>
> |Estimators|  Step 3000  ($\text{W}_2\downarrow$)| Step 4000 ($\text{W}_2\downarrow$)| Step 5000  ($\text{W}_2\downarrow$)| Step 6000($\text{W}_2\downarrow$) | Step 8000 ($\text{W}_2\downarrow$)|Times (s)|
> |---------------|----------|----------|-----------|---------|------------|----|
> |PW L=3| $385.33538\pm4.76$ | $214.81157\pm4.6$|$92.94936\pm3.15$|$22.02469\pm2.23$|$0.00614\pm0.0$| $36.31\pm0.82$
>
> We see that the performance of PW is not as good as SW in Table 1 in both computation and quantitative scores.
>
> Overall, we believe the comparative performance of distances depends on the applications i.e., each distance has its preferred applications. In our work, we focus on SW only i.e., focusing on the expectation with respect to the uniform slicing distribution. Since constructing control variates for SW is a new area of research, we believe the extension to other variants of SW is worth a careful future investigation.
>
> We are happy to discuss more if the reviewer is not satisfied with our response.
>
> Best regards,

---

> > ### Comment · Reviewer_yEdx · 2023-11-20
> >
> > Thanks for the detailed answer. I have raised my score to 6.

---

> > > ### Author Response · Authors · 2023-11-20
> > > **Response to Reviewer yEdx**
> > >
> > > Thank you for raising the score to 6. Please feel free to ask if you have additional questions. We would like to show our appreciation again for your constructive comments.
> > >
> > > Best regards,

---

### Official Review · Reviewer_mJzC · 2023-10-31

**Soundness:** 3 good
**Presentation:** 3 good
**Contribution:** 2 fair
**Rating:** 6
**Confidence:** 4

**Summary:**

Wasserstein (W) distance plays an increasingly preponderant role in many machine learning pipelines, since its ability to capture geometric features of the objects at hand. However, it suffers from a heavy computational cost, $O(n^3\log(n))$ where $n$ is the number of supports of the probability measures. To overcome this computational bottleneck, sliced Wasserstein distance (SWD) stands as an alternative to Wasserstein distance. SWD is based on slicing the origin measures  $\mu$ and $\nu$ by projecting them on a direction $\theta$ of the unit-hypersphere and then calculating an expectation (average) of the 1-dimensional Wasserstein distances between the projected measures $\mu_\theta$ and $\nu_\theta$. This expectation is calculated using Monte Carlo (MC) integration to estimate SWD distance. It is known that the error of the MC approximation is $O\big({L}^{-1/2} \times \text{Variance}[W_p^p(\mu_\theta, \nu_\theta)\big]\big)$.

This paper proposes a computationally efficient control of the term $\text{Variance}[W_p^p(\mu_\theta, \nu_\theta)\big]$ based on the control variates from the literature on variance reduction. The idea behind this is to find left and right control variates estimators, which are Gaussian control approximations of $\mu_\theta$ and  \nu_\theta$ with a low variance of their Wasserstein distances.  These estimators share the same computational complexity and memory complexity as the conventional estimator of SWD (vanilla MC estimator).

**Strengths:**

- Proposing a novel control of the variance term in the projection complexity of SWD that leads to a computationally efficient estimation of SWD.
- The estimators are based on the control variate from the literature of reduction variance, which seems interesting to bridge the OT metrics with reduction variance.
- Extensive experiments on comparing probability measures over images and point-clouds, point clouds gradient flows, and generative modeling. The left and right estimators have computational and performance gains over the conventional MC approximation of SWD.
- The presentation of the paper is easy to follow. I checked the proofs and they sound good to me.
- The code is attached and the experimental protocols are well explained (in the main + supplementary), which guarantees the reproducibility of the experiments.

**Weaknesses:**

- In Definition 1, the expectation of the controlled projected one-dimensional Wasserstein distance is non-nonegative (unbiased estimator of SWD). But what about the variable itself, $Z(\theta, \mu, \nu)$, is it also non-negative?
- I'm wondering about the utility of Proposition 4, which states the left control estimator is equivalent to considering a control variate with respect to the 2-Wasserstein distance between a projection of Gaussian approximations of the origin measures. I'm thinking about the following scheme: we first calculate these Gaussian approximations of the origin probability measures then an SWD between these approximations and the controlled SWD is given by: $|\hat{SWD}(\text{origin measures}) - \hat {SWD}(\text{Gaussian Approximation of the origin measures})|$ without adding the factor $\gamma$.
I don't know it may highlight the importance of $\gamma$ to get more ``tight`` control variate. Any comment(s) will be appreciated for this point.
- The most related previous work is the Gaussian approximation for the SWD_2 (Nadjahi et al; n NeurIPS'21). Can we expect a fast rate of convergence of the controlled variate estimator, e.g. LCV-SWD to the true Sliced Wasserstein distance? (In Nadjahi et al; n NeurIPS'21, the rate is $O(d^{-1/8})$, see Corollary 1 therein, which is too slow for $d \gg1$.)

**Questions:**

### Minor Typos
- Page 2 (last line): there is an extra $F_\nu$.
- Page 4: the term $\gamma^2 \text{Var}[C(\theta)]$ is extra in the derivative $f'(\gamma)$.
- Page 4: "... has a correlation with $W(\theta)$" --> " ... has a correlation with $W(\theta; \mu, \nu)$"
- Page 5: in Definition 3 "... i.e.$\mathcal{N}$ ..." missing space
- Page 12: there is no sign "-" on the derivative of $f(m_1, \sigma^2)$ wrt $m_1$.
- Page 13:  the notation $[[\cdot]]$ is not defined.

---

> ### Author Response · Authors · 2023-11-16
> **Response to Reviewer mJzC (Part 1)**
>
> We would like to express our gratitude for the time and feedback from the reviewer. We would like to extend the discussion as follows:
>
> **Q10**:In Definition 1, the expectation of the controlled projected one-dimensional Wasserstein distance is non-nonegative (unbiased estimator of SWD). But what about the variable itself, $Z(\theta;\mu,\nu)$
> , is it also non-negative?
>
> **A10**: Thank you for your insightful question. The controlled variable $Z(\theta;\mu,\nu)$ can be negative, however, $\mathbb{E}[Z(\theta;\mu,\nu)]$ is still non-negative since it is the SW distance. Moreover,   $\mathbb{E}[C(\theta)-B] =0$ which makes its estimators with Monte Carlo samples might be closed to 0 with high probability. Therefore, the Monte Carlo estimators of $\mathbb{E}[Z(\theta;\mu,\nu)]$ might be non-negative with high probability.
>
> **Q11**: I'm wondering about the utility of Proposition 4, which states the left control estimator is equivalent to considering a control variate with respect to the 2-Wasserstein distance between a projection of Gaussian approximations of the origin measures. I'm thinking about the following scheme: we first calculate these Gaussian approximations of the origin probability measures then an SWD between these approximations and the controlled SWD is given by:
> |$\hat{SWD}$(origin measures)−$\hat{SWD}$(Gaussian Approximation of the origin measures)|
>  without adding the factor  $\gamma$
> . I don't know it may highlight the importance of $\gamma$
>  to get more tight control variate. Any comment(s) will be appreciated for this point.
>
> **A11**: First we would like to recall that SW-2 distance only has a closed-form when two measures are location-scale measures. For multivariate Gaussian with a diagonal covariance matrix and full covariance matrix, SW-2 is still approximated by the MC estimator.
>
> In terms of approximation, approximating a high-dimensional measure by a Gaussian is likely to create more error than approximating an uni-dimensional measure by a Gaussian. Moreover, according to some central limit theorem [R1], the projected one-dimensional measure converges in distribution to a Gaussian measure. Therefore, we believe it is more appealing to do an approximation after the projection.
>
> For the importance of $\gamma^\star$, we assume that you are referring to the difference estimator.  In more detail, the estimator with $\gamma^\star$ is called the regression estimator. When $\gamma=1$, the corresponding estimator is named the difference estimator. While the difference estimator is simpler and does not need to deal with estimating $\gamma^\star$, the performance of the difference estimator depends strongly on the approximation error between the control variate and the interested integrand. For a bad approximation, the resulting difference estimator can lead to worse variance. In contrast, the regression estimator always leads to lower variance with a good approximation of $\gamma^\star$ i.e., can be guaranteed by increasing the number of MC samples. We would like to refer to Page 29, Chapter 8 in [R2] for a more detailed discussion.
>
>
> [R1] Typical distributions of linear functionals in finite dimensional spaces of high dimension.
>
> [R2] Monte Carlo theory, methods and examples

---

> > ### Author Response · Authors · 2023-11-16
> > **Response to Reviewer mJzC (Part 2)**
> >
> > **Q12**:  The most related previous work is the Gaussian approximation for the SWD_2 (Nadjahi et al; n NeurIPS'21). Can we expect a fast rate of convergence of the controlled variate estimator, e.g. LCV-SWD to the true Sliced Wasserstein distance? (In Nadjahi et al; n NeurIPS'21, the rate is $\mathcal{O}(d^{-1/8}$
> > , see Corollary 1 therein, which is too slow for $d≫1$
> > .
> >
> > **A12**: Thank you for a detailed comment. First, we would like to highlight the difference between our work and [R3]. The work [R3] focuses on forming a deterministic approximation of SW based on the concentration of the random projection when the number of dimensions increases. As mentioned by the reviewer, the approximation error of such approximation reduces very slowly in terms of $d$. Therefore, the proposed approximation in  [R3] has not been widely used in practice. In addition, deep learning application requires the computation of gradient. A deterministic approximation often does not lead to good results compared to a stochastic estimation.
> >
> > In contrast, the approximation in our work is only for constructing the control variates. If the approximation is not good, it does not affect the performance of the Monte Carlo estimation i.e., the variance stays the same without reducing. Of course, a good approximation will extract more statistical relevance between the control variate and the integrand. However, as discussed with Reviewer dx24 in Q9.  We need control variates that have a tractable expectation and fast computation (at most $\mathcal{O}(Ln\log n +Ldn )$). Hence, the current solution is to construct them as a combination of the sample mean (in terms of support) and sample variance.
> >
> > Our control variates are constructed as a lower bound and an upper bound of Wasserstein-2 between two Gaussians that are obtained from minimizing KL divergence / maximum likelihood/ moment matching. Therefore, to investigate the error, we need to understand the asymptotic behavior of the MLE first, then understand the gap between the lower-bound and upper bound to the true distance. From that, we might obtain the asymptotic result for the approximation of our control variate. Unfortunately, that process is quite challenging at the moment, hence, we leave that to future work. Nevertheless, as discussed in the above paragraph, the control variate must not be very accurate to help reduce variance.
> >
> > [R3] Fast Approximation of the Sliced-Wasserstein Distance Using Concentration of Random Projections
> >
> > **Q13**: On typos
> >
> > **A13**: We have addressed all mentioned typos in the revision in blue.

---

### Official Review · Reviewer_dx24 · 2023-11-01

**Soundness:** 2 fair
**Presentation:** 3 good
**Contribution:** 2 fair
**Rating:** 6
**Confidence:** 3

**Summary:**

The paper proposes novel Monte Carlo estimators for the sliced Wasserstein distance (SW) using the control variates principle. The proposed estimators have reduced variance, and the computation costs remain the same. In addition, the paper provides various experiments and applications, which illustrate that in practice the estimators do outperform naive SW estimator, and can be applied to various tasks including gradient flow and generative modeling.

**Strengths:**

The paper is overall clear and well presented, and the results are original and novel to the knowledge of the reviewer. The strengths of the paper includes:

1. The statistical aspects of estimator of the SW is mostly studied from the perspective of the marginal populations, and this paper sheds new light on the MC side of estimation of the SW, from a variance reduction point of view. This can be of practical interests.
2. The construction of the estimators makes extensive use of properties of Wasserstein distance, especially in the 1 dimensional case. The balance of computational tractability and statistical relevance seems an interesting aspect, especially for SW as it utilizes external randomness.
3. The paper provides extensive experiments, which seem sufficient for justifying the practicality of the proposed estimators.

**Weaknesses:**

Some weaknesses:

1. A major confusion is proposition 1, in which the paper claims to minimize KL divergence between discrete distribution and a Gaussian distribution. However, to the best of the reviewer's knowledge, there's no way to obtain finite value for KL between discrete distribution and continuous distribution, even under the most generalized setting with Radon–Nikodym derivative. The reviewer acknowledge that this potential flaw does not defeat the purpose of identifying the Gaussian proxy using information of $\mu,\nu$ (e.g. as an alternative, through moment matching), but urge the authors to revise this part accordingly.
2. The usage of 'upper/lower bounds' seems to lack justification, as they are quite different from the Gaussian approach the paper proposes, in terms of how much correlation is changed/lost. In addition, though $\mathbb{E}[\sigma_1(\theta;\mu)\sigma_2(\theta;\nu)]$ does not have a closed form, it's unclear why it can't be estimated by MC, as is done throughout the paper. (A personal thought: from a statistical perspective, in order to introduce correlation it might be possible to use $\sigma_1^2(\theta;\mu)\sigma_2^2(\theta;\nu)$, if it is at all possible to address.)
3. The treatment of the resulting estimator from control variates is rough. In particular, the following items need proper discussion: unbiasedness (the biasedness after taking p-root does not justify the biasedness of proposed estimator which is without a root), actual variance of the proposed estimators (which is not the control variate estimator but a further estimator of it), and how good the proposed estimators of coefficients are (e.g. $\hat{\gamma^*}$).

**Questions:**

Please see above (section Weaknesses) for details. The reviewer is confident that these can be clarified.

To make the paper stronger, the reviewer suggest investigating the balance between the computational tractability (e.g. using $m_1,\sigma_1$) and statistical relevance, as it seems that the easier control variate is to be computed, the less information it provides. Theoretical or empirical evidence would be illuminating.

---

> ### Author Response · Authors · 2023-11-16
> **Response to Reviewer dx24 (Part 1)**
>
> We appreciate the time and feedback from the reviewer. We would like to address the concerns from the reviewer as follow:
>
> **Q6**: A major confusion is proposition 1, in which the paper claims to minimize KL divergence between discrete distribution and a Gaussian distribution. However, to the best of the reviewer's knowledge, there's no way to obtain finite value for KL between discrete distribution and continuous distribution, even under the most generalized setting with Radon–Nikodym derivative. The reviewer acknowledge that this potential flaw does not defeat the purpose of identifying the Gaussian proxy using information of $\mu$ and $\nu$(e.g. as an alternative, through moment matching), but urge the authors to revise this part accordingly.
>
> **A6**:  Thank you for your insightful comment. In the paper, we define KL divergence using the convention that $0\log 0=0$ i.e., $\lim_{x\to 0}x \log p(x) = 0$ as defining entropy in information theory literature. As commented by the reviewer, the main goal of defining the optimization problem is to find a Gaussian approximation of two discrete measures i.e., using the average of supports as the mean, and using the average square-L2 between supports and the mean as the variance. The formulation can also interpreted via Moment matching as suggested. We would like to recall that the Gaussian approximation can use any approximation schemes and the approximation in the paper is only a well-known type of approximation.
>
> We have added this discussion to the paper in Section 3  in blue color. If the reviewer is not satisfied with the explanation with convention  $0\log 0=0$, we will directly state the approximation formula instead of defining the optimization problem with the KL divergence.
>
> [R1] Information, Physics and Computation, Chapter 1, page 3.
>
> **Q7**:The usage of 'upper/lower bounds' seems to lack justification, as they are quite different from the Gaussian approach the paper proposes, in terms of how much correlation is changed/lost. In addition, though $\mathbb{E}_{\sigma_1(\theta;\mu)\sigma_2(\theta;\nu)}$
>  does not have a closed form, it's unclear why it can't be estimated by MC, as is done throughout the paper. (A personal thought: from a statistical perspective, in order to introduce correlation it might be possible to use $\mathbb{E}[\sigma_1^2(\theta;\mu)\sigma_2^2(\theta;\nu)]$
> , if it is at all possible to address.)
>
>
> **A7**: Thank you for your insightful comments. The main goal of the paper is to create a first attempt to connect the control variate literature and the sliced Wasserstein literature. To use control variate, we need to have a measurable function that has a tractable expectation and has a correlation with the original function. Motivated by the well-known result of the closed-form solution of Wasserstein distance between two Gaussians, we construct the control variate in the presented way. We would like to recall that the proposed control variates might not be optimal in the sense of reducing variance. They are designed as natural and computationally efficient control variates.
>
> The reason why we do not estimate $\mathbb{E}[\sigma_1(\theta;\mu),\sigma_2(\theta;\nu)]$ via MC methods is because we do not want to create nested MC problems i.e., the expectation of the control variate is only intractable. In principle, we could use MC to estimate such quantity. However, it might lead to a little conflict with the original motivation and the conventional framework of using control variate. We have run an additional experiment in gradient flow with the control variate including estimating $\mathbb{E}[\sigma_1(\theta;\mu),\sigma_2(\theta;\nu)]$ via MC method.  We have the following result:
>
> |Estimators|  Step 3000  ($\text{W}_2\downarrow$)| Step 4000 ($\text{W}_2\downarrow$)| Step 5000  ($\text{W}_2\downarrow$)| Step 6000($\text{W}_2\downarrow$) | Step 8000 ($\text{W}_2\downarrow$)|
> |---------------|----------|----------|-----------|---------|------------|
> L=10| $304.04405\pm0.27$ | $137.11704\pm0.14$|$35.75328\pm0.11$|$0.13183\pm0.01$|$2e-05\pm0.0$|
> L=100| $300.91719\pm0.01$|$134.0464\pm0.0$|$33.79072\pm0.0$|$0.06195\pm0.0$|$2e-05\pm0.0$|
>
> Compared to Table 1 in the paper, despite being slightly better than SW without a control variate, this estimator is worse than the proposed control variates in the paper. As discussed, it might be due to the nested MC problem.
>
> For using $\mathbb{E}[\sigma_1^2(\theta;\mu),\sigma_2^2(\theta;\nu)]$, to the best of our knowledge, the expectation is also intractable since it leads to a polynomial function of $\theta$ with the degree 4. Overall, we believe that the construction of the control variates is worth further investigation.

---

> > ### Author Response · Authors · 2023-11-16
> > **Response to Reviewer dx24 (Part2 )**
> >
> > **Q8**: The treatment of the resulting estimator from control variates is rough. In particular, the following items need proper discussion: unbiasedness (the biasedness after taking p-root does not justify the biasedness of proposed estimator which is without a root), actual variance of the proposed estimators (which is not the control variate estimator but a further estimator of it), and how good the proposed estimators of coefficients are (e.g. $\hat{\gamma^\star}$)
> >
> > **A8**: Thank you for your insightful question. For unbiasedness, we can use a separate set of  Monte Carlo samples to estimate $\gamma^\star$ as discussed in Remark 1. However, that estimator costs double the computation of SW.  Moreover, although the reused Monte Carlo samples scheme is biased, it is widely shown in practice that the bias is usually small (page 30 Chapter 8 in [R2]).
> >
> > For the theoretical understanding, as mentioned by Reviewer pTc9 in Q4, the literature for the approximation error with control variate is quite limited for intractable optimal coefficient. Therefore, control variate has been mainly seen as a practical way to improve the approximation error via reducing the variance of estimators.
> >
> > [R2] Monte Carlo theory, methods and examples
> >
> >
> > **Q9**: To make the paper stronger, the reviewer suggests investigating the balance between the computational tractability (e.g. using ($m_1,\sigma_1$) and statistical relevance, as it seems that the easier control variate is to be computed, the less information it provides. Theoretical or empirical evidence would be illuminating.
> >
> > **A9**: We agree that the easier the control variate is to be computed, the more likely that it provides less information about the intergrand. Nevertheless, the main benefit of using SW is because of its fast computation i.e., $\mathcal{O}(Ln\log n +Ldn )$ for $n$ is the number of supports, $d$ is the number of dimensions, and $L$ is the number of Monte Carlo samples. As a result, the computation of the control variate should have the computational complexity at most $\mathcal{O}(Ln\log n +Ldn)$. Combined with the requirement of tractable expectation, designing the control variate for SW is quite challenging. At the moment, control variates, which satisfy both criteria, can be constructed as a combination of the sample mean (in terms of supports) and sample variance.  There may exist other control variate form that is more expressive while being computationally efficient. However, the construction for such control variate is not trivial at the current state. We would like to recall that the paper is the first attempt to bridge the literature of variance reduction (e.g., control variate) with the SW literature. The solution in the paper might not be optimal in both computation and performance, however, it is the most natural attempt to bridge the two literatures.

---

> > > ### Comment · Reviewer_dx24 · 2023-11-20
> > > **Thanks for the response**
> > >
> > > I thank the authors for the response, which addresses all my questions. I thus leave my rating unchanged.
> > >
> > > A side note: Taking $0\log 0$ to be 0 is standard and widely used, but KL between discrete and continuous distributions $P,Q$ requires $\frac{dP}{dQ}$, which is $\infty$ in the Radon–Nikodym sense on the discrete support of $P$, and the $0\log0$ convention is only applied outside the support of $P$. Though not hindering the purpose of the paper, the reviewer still insists that the usage of KL divergence being revised, as the infinity arises naturally as part of consistency of KL diergence. For introductory reference please see [1].
> > >
> > > [1] https://math.stackexchange.com/questions/164744/kl-divergence-between-bernoulli-distribution-with-parameter-p-and-gaussian-dis?noredirect=1&lq=1

---

> > > > ### Author Response · Authors · 2023-11-21
> > > > **Response to Reviewer dx24**
> > > >
> > > > We would like to thank the reviewer again for the additional constructive feedback,
> > > >
> > > > We have removed all KL divergence definitions in the paper. We directly state the optimization as a maximum weighted log-Gaussian-likelihood objective :
> > > >
> > > > $$m_1(\theta;\mu),\sigma_1^2(\theta;\mu) =argmax_{m_1,\sigma_1^2}\left[ \sum_{i=1}^n \alpha_i\log \left(\frac{1}{\sqrt{2\pi \sigma_1^2}} \exp \left(-\frac{1}{2 \sigma_1^2} (\theta^\top x_i - m_1)^2 \right)\right)\right].$$
> > > >
> > > > It would be great if the reviewer could give us comments on the revision.
> > > >
> > > > Best regards,

---

> > > > > ### Comment · Reviewer_dx24 · 2023-11-21
> > > > > **On revision**
> > > > >
> > > > > The revised claims all look good.

---

> > > > > > ### Author Response · Authors · 2023-11-22
> > > > > > **Response to Reviewer dx24**
> > > > > >
> > > > > > Thank you very much for your feedback!

---

### Official Review · Reviewer_pTc9 · 2023-11-02

**Soundness:** 3 good
**Presentation:** 2 fair
**Contribution:** 3 good
**Rating:** 8
**Confidence:** 4

**Summary:**

Developing efficient algorithms to compute slided Wasserstein distance is an important problem in OT. The authors proposed to use control variates, a special variance reduction algorithm, to compute the distance reliably. Compared with standard Monte Carlo sampling, it has a much smaller variance, as observed empirically. The authors carefully designed the control variates, which plays a crucial role in the performance of variance reduction. The computational complexity is also analyzed, and a comprehensive numerical study is performed to show the superior performance of their proposed algorithm.

**Strengths:**

- Although the idea of control Variate is simple and straightforward, the construction of the control variate estimator is highly non-trivial. The authors find a good estimator based on the closed-form expression of the OT distance between two fitted Gaussians. This idea is useful.
- I agree that it is intractable to evaluate $\mathbb{E}[W_2^2(\mathcal{N}(m_1(\theta;\mu), \sigma_1^2(\theta;\mu)), \mathcal{N}(m_2(\theta;\mu), \sigma_2^2(\theta;\mu)))]$, and the authors proposed lower and upper bounds on this unknown quantity around Proposition 3, which is valid.
- The computational complexity for two control variate-based sliced Wasserstein distance is analyzed, which showcases their computational efficiency.
- It is good to see how to apply the authors' algorithm to other variants of sliced Wasserstien distance, as pointed at the end of Section 3.
- It is also interesting to see how to apply the control variate to compute the gradient of sliced Wasserstein distance, as pointed at the end of Section 3.
- Numerical study on three different applications is solid.

**Weaknesses:**

While I appreciate and understand the authors' main idea, some parts of writing can be improved:
- It is good that the authors provided strong motivations for using control covariates in Section 3.1, paragraph 2, the wiring in this part is poor. I suggest the authors follow the writing of [A. Shapiro 2021, Section 5.5.2] to re-write this part.

Ref: Shapiro A, Dentcheva D, Ruszczynski A. Lectures on stochastic programming: modeling and theory[M]. Society for Industrial and Applied Mathematics, 2021.

- For the paragraph "Constructing Control Variates" in Section 3.2, the authors should omit the detailed deviation of the closed-form solution of $W_2^2(\mathcal{N}(m_1(\theta;\mu), \sigma_1^2(\theta;\mu)), \mathcal{N}(m_2(\theta;\mu), \sigma_2^2(\theta;\mu)))$. It is fine to present only the final simplified expression.
- For the paragraph below Proposition 1, the description for computing the sliced Wasserstein between continuous distributions is brief and confusing. The authors may consider describe the algorithms in detail and present this part in an extra Appendix instead.
- I feel the theoretical analysis of the variance-reduced estimator is not enough. But when I look at related literature, there is little theoretical guarantees on this part. So I think it is fine regarding the theoretical contribution part.

**Questions:**

I think the authors also miss an important application on computing sliced Wasserstein (SW) distance. Since the SW distance can be used to quantify the difference between distributions, it can be used for non-parametric two-sample testing, i.e., given samples from two distributions $\mu$ and $\nu$, to determine either $H_0: \mu=\nu$ or $H_1: \mu\ne\nu$. Related literature [Wang et al. 2021] used the projected Wasserstein distance (seeks the one-dimensional projector that maximizes OT distance, instead of finding the averaged OT distance among all one-dimensional projectors) for two-sample testing, but it can be naturally extended for SW distance. The authors can use their variance reduction technique to finish this task with superior computational efficiency.

Ref: Wang J, Gao R, Xie Y. Two-sample test using projected wasserstein distance[C]//2021 IEEE International Symposium on Information Theory (ISIT). IEEE, 2021: 3320-3325.

---

> ### Author Response · Authors · 2023-11-16
> **Response to Reviewer pTc9 (Part 1)**
>
> We would like to thank the reviewer for the time and feedback. We would like to answer questions from the reviewer as follow:
>
> **Q1**: It is good that the authors provided strong motivations for using control covariates in Section 3.1, paragraph 2, the wiring in this part is poor. I suggest the authors follow the writing of [A. Shapiro 2021, Section 5.5.2] to re-write this part.
>
> **A1**: Thank you for pointing out. We realized that there are two definitions of the controlled variable $Z(\theta;\mu,\nu)$ i.e., one with $\gamma$ and one with optimal $\gamma$. We have rewritten Section 3.1 paragraph 2 in blue color based on Section 5.5.2 in [R1]. We would be grateful if the reviewer could give comments on the rewritten paragraph.
>
> [R1] Lectures on stochastic programming: modeling and theory
>
> **Q2**: For the paragraph "Constructing Control Variates" in Section 3.2, the authors should omit the detailed deviation of the closed-form solution of  $ \text{W}_2^2(\mathcal{N}(m_1(\theta;\mu),\sigma_1^2(\theta;\mu)),\mathcal{N}(m_2(\theta;\nu),\sigma_2^2(\theta;\nu)))$. It is fine to present only the final simplified expression.
>
> **A2**:  Thank you for your suggestion. We have rewritten Section 3.2 based on your suggestion in blue color.
>
> **Q3**: For the paragraph below Proposition 1, the description for computing the sliced Wasserstein between continuous distributions is brief and confusing. The authors may consider describing the algorithms in detail and present this part in an extra Appendix instead.
>
> **A3**: Thank you for the suggestion. We would like to extend the discussion as follows:
>
> Laplace Approximation. We are interested in approximating a continuous distribution $\mu$ by an Gaussian $\mathcal{N}(m,\sigma^2)$. We assume that we know the pdf of $\mu$ which is doubly differentiable and referred to as $f(x)$. We first find $m$ such that $f'(m)=0$. After, we use the second-order Taylor expansion of $\log f(x)$ around $m$:
> $$
>     \log f(x)\approx \log f(m) - \frac{1}{2\sigma^2}(x-m)^2,
> $$
> where the first order  does not appear since $f'(m)=0$ and  $\sigma^2 = \frac{-1}{\frac{d^2}{dx^2} \log f(x) \big{|}_{x=m}}$.
>
>
> Sampling-based approximation. We are interested in approximate a continuous distribution $\mu$ by an Gaussian $\mathcal{N}(m,\sigma^2)$. Here, we assume that we do not know the pdf of $\mu$, however, we can sample from $X_1,\ldots, X_n \overset{i.i.d}{\sim}\mu$.   Therefore, we can approximate $\mu$ by $\mathcal{N}(m,\sigma^2)$ with $m=\frac{1}{n}\sum_{i=1}^n X_i$ and $\sigma^2 =  \frac{1}{n}\sum_{i=1}^n (X_i-m)^2$ which is equivalent to doing maximum likelihood estimate, moment matching, and doing optimization in Equation 11 with the proxy empirical measure $\mu_n=\frac{1}{n}\sum_{i=1}^n \delta_{X_i}$.
> }
>
> We have added the discussion in Appendix A in the revision in blue. We would be grateful if the reviewer could give us some comments.
>
>
> **Q4**: I feel the theoretical analysis of the variance-reduced estimator is not enough. But when I look at related literature, there is little theoretical guarantees on this part. So I think it is fine regarding the theoretical contribution part.
>
> **A4**: Thank you for your detailed comment. For a tractable form of the optimal coefficient $\gamma$ i.e., $Z(\theta;\mu,\nu,C(\theta)) in Definition 1, using control variates always leads to a lower variance (or at least the same), then lead to a lower approximation error in L1 norm. Nevertheless, the optimal coefficient is often intractable, hence, Monte Carlo samples are used to approximate the optimal coefficients. By using Monte Carlo samples, it is challenging to investigate the approximation error of the empirical estimator since it involves the ratio and the product of estimators of SW, covariance, and variances. As mentioned by the reviewer, the theoretical understanding of the approximation error is still an open area in the control variate literature even with some simple forms of the integrand. In our case, the integrand is SW which is a function on the unit-hypersphere (a manifold) and involves an optimal transport problem. Therefore, we would like to leave this investigation to the future work. Overall, we believe control variate can be mainly seen as a practical method to improve the performance of estimators in practice which has been shown in the paper with various applications.

---

> > ### Author Response · Authors · 2023-11-16
> > **Response to Reviewer pTc9 (Part 2)**
> >
> > **Q5**: I think the authors also miss an important application on computing sliced Wasserstein (SW) distance. Since the SW distance can be used to quantify the difference between distributions, it can be used for non-parametric two-sample testing, i.e., given samples from two distributions $\mu$ and $\nu$, $H_0:\mu=\nu$ or $H_1:\mu\neq \nu$. Related literature [Wang et al. 2021] used the projected Wasserstein distance (seeks the one-dimensional projector that maximizes OT distance, instead of finding the averaged OT distance among all one-dimensional projectors) for two-sample testing, but it can be naturally extended for SW distance. The authors can use their variance reduction technique to finish this task with superior computational efficiency.
> >
> > **A5**: Thank you for your helpful reference. We have conducted experiments on a non-parametric two-sample test in Appendix B in the revision. We follow the setting in your mentioned work [R2]. Instead of using projected Wasserstein distance, we use SW as the testing distance. We use the setting to show that our proposed control variate estimators lead to a stronger-power test.
> >
> > We use SW, LCV-SW, and UCV-SW in the Gausian hypothesis-testing setting.  We observe that LCV-SW and UCV-SW give slightly stronger power for the test than SW without control variates. For Type I Error, UCV-SW is better than SW and LCV-SW is the same as SW.  We refer the reviewer to Appendix B for more details of experiments.
> >
> > [R2]  Two-sample test using projected Wasserstein distance

---

> > > ### Comment · Reviewer_pTc9 · 2023-11-16
> > > **Score after rebuttal**
> > >
> > > I feel the reviewer successfully addressed all of my concerns. I decide to raise my score to 8.

---

> > > > ### Author Response · Authors · 2023-11-16
> > > > **Response to Reviewer pTc9**
> > > >
> > > > Thank you for raising the score to 8. Please feel free to ask if you have additional questions. We would like to thank you again for giving us a lot of constructive comments.
> > > >
> > > > Best regards,

---

### Comment · Reviewer_mJzC · 2023-11-21

I thank the authors for the detailed responses. I keep my score unchanged.

---

> ### Author Response · Authors · 2023-11-21
> **Response to Reviewer mJzC**
>
> We would like to thank you again for your time and constructive feedback!
>
> Best regards,

---

### Meta-Review · Area_Chair_7UGr · 2023-12-06

**Metareview:**

The paper should be accepted due to its clear and well-presented content, providing a non-trivial contribution to the control variate estimator. It offers an elegant solution accompanied by a thorough theoretical analysis. Additionally, the paper stands out for its solid and extensive numerical study, with experimental protocols that are well explained, showcasing the depth and rigor of the research. This combination of clarity, theoretical soundness, and practical robustness in experimentation marks a significant advancement in the field.

**Justification For Why Not Higher Score:**

It is the best paper in my batch. The grade are not very good, but the reviewers might have been a bit harsh. I still recommend only poster, but it is borderline to spotlight.

**Justification For Why Not Lower Score:**

The paper is clearly above the threshold.

---

### Decision · Program_Chairs · 2024-01-16

Accept (poster)